# Stylized Offline Reinforcement Learning: Extracting Diverse High-Quality Behaviors from Heterogeneous Datasets

**Yihuan Mao[1], Chengjie Wu[1], Xi Chen[1], Hao Hu[1], Ji Jiang[2], Tianze Zhou[2], Tangjie Lv[2], Changjie Fan[2], Zhipeng Hu[2], Yi Wu[1], Yujing Hu[2]\*, Chongjie Zhang[3]\***
[1] Institute for Interdisciplinary Information Sciences, Tsinghua University, China
[2] Fuxi AI Lab, NetEase, China,   [3] Washington University in St. Louis
`{maoyh20,wucj19,huh22}@mails.tsinghua.edu.cn`
`{pcchenxi,jxwuyi}@tsinghua.edu.cn`,   `chongjie@wustl.edu`
`{jiangji,zhoutianze,hzlvtangjie,fanchangjie,zphu,huyujing}@corp.`
`netease.com`

## Abstract

Previous literature on policy diversity in reinforcement learning (RL) either focuses on the online setting or ignores the policy performance. In contrast, offline RL, which aims to learn high-quality policies from batched data, has yet to fully leverage the intrinsic diversity of the offline dataset. Addressing this dichotomy and aiming to balance quality and diversity poses a significant challenge to extant methodologies. This paper introduces a novel approach, termed Stylized Offline RL (SORL), which is designed to extract high-performing, stylistically diverse policies from a dataset characterized by distinct behavioral patterns. Drawing inspiration from the venerable Expectation-Maximization (EM) algorithm, SORL innovatively alternates between policy learning and trajectory clustering, a mechanism that promotes policy diversification. To further augment policy performance, we introduce advantage-weighted style learning into the SORL framework. Experimental evaluations across multiple environments demonstrate the significant superiority of SORL over previous methods in extracting high-quality policies with diverse behaviors. A case in point is that SORL successfully learns strong policies with markedly distinct playing patterns from a real-world human dataset of a popular basketball video game "Dunk City Dynasty."

## 1 Introduction

Learning to accomplish a task with diverse behaviors, also known as quality-diversity optimization, is an emerging area within stochastic optimization research (Pugh et al., 2016; Cully & Demiris, 2018; Mouret & Clune, 2015). It aims to generate a diverse set of solutions that maximize a given objective function and is especially valuable in applications involving human interactions, such as games (Shen et al., 2021) and autonomous driving (Araujo et al., 2023). For instance, in online games, deploying AI bots with varied motion styles can enrich the gaming environment and enhance player engagement. Similarly, in autonomous driving, offering multiple solutions can cater to users with different preferences. Additionally, in opponent modeling (Yu et al., 2022), high-quality diverse opponents that resemble real opponents can significantly improve the performance of the learned policy. However, a significant limitation is that most studies in this domain heavily rely on extensive online interactions (Nilsson & Cully, 2021; Pierrot et al., 2022), leading to high costs and imposing constraints on systems with restricted online access. Recent advances in offline RL present a promising direction, allowing for policy learning from pre-collected datasets without further interactions (Fujimoto et al., 2019; Kumar et al., 2020; Kostrikov et al., 2021). However, they often prioritize policy quality, sidelining the inherent diversity within the dataset. In this paper, we bridge this gap by targeting both diversity and high-quality policy learning from offline datasets.

---

\*Equal advising.
  Code is at `https://github.com/cedesu/SORL`.

The central challenge lies in how to optimize the performance of the policies while ensuring that their behaviors are as distinguishable as possible. Balancing this in an offline setting becomes problematic due to the inherent characteristics of offline datasets. Typically, datasets capturing diverse behaviors are heterogeneous, gathered from multiple sources, leading to an action distribution with multiple modes and inconsistent data quality (Chen et al., 2022; Li et al., 2017). Studies focus on learning diverse behaviors that can capture the multi-modality of the action distribution. However, they often employ a diversity objective that is task-agnostic (Eysenbach et al., 2019; Masood & Doshi-Velez, 2019). In offline settings, this task-agnostic objective can lead to policies that perform exceedingly poorly due to inconsistent data quality. Conversely, current offline RL methods prioritize task performance maximization. To mitigate overestimation issues, a conservative constraint is often imposed on the policy, ensuring that its distribution aligns closely with the dataset (Fujimoto et al., 2019; Kumar et al., 2020; Kostrikov et al., 2021). Amplifying this conservativeness might retain some multi-modality of the dataset. However, it does not offer control over how distinguishable the policy should be during training.

To address this challenge, we introduce Stylized Offline Reinforcement Learning (SORL), a two-step framework designed to derive diverse and high-quality policies from a heterogeneous offline dataset. In the first step, we perform a style clustering. Drawing inspiration from the venerable Expectation-Maximization (EM) algorithm, we classify the trajectories from the heterogeneous dataset into clusters where each represents a distinct and dominant motion style. In the second step, we employ advantage-weighted style learning, an offline RL method with a novel objective considering both performance and diversity. Here we train a set of policies to maximize the task performance, with each policy specifically constrained to the action distribution of a particular style identified in the first step. Unlike other offline RL methods that constrain the policy to the entire dataset without differentiating between types of motions, our approach effectively extracts stylistically diverse behavior in the dataset that can be characterized by distinct behavioral patterns. In contrast to diverse RL methods, we achieve high-performing policies that are in-distribution with respect to the dataset, yet less influenced by the low-quality samples.

We evaluate SORL across various environments and offline datasets. These include a didactic game with a hand-crafted dataset, a set of Atari games using open-sourced human data, and the popular basketball video game "Dunk City Dynasty" with data recorded from online players. We compare SORL against the offline versions of two baseline methods that focus on learning diverse behavior from multi-modal datasets. The experimental results demonstrate the significant superiority of SORL over the baseline methods in achieving policies with higher performance while maintaining distinguishable behavior patterns. In summary, the contributions of this paper are as follows:

1. We introduce SORL, a novel framework that addresses the limitations of both diverse RL and offline RL methods by incorporating both quality and diversity into the optimization objective.

2. We provide comprehensive evaluations of SORL across diverse environments using datasets recorded from humans, showcasing its capability to extract high-performing, stylistically diverse policies from heterogeneous offline datasets.

## 2 RELATED WORK

**Offline RL and Imitation Learning**   Offline reinforcement learning (RL) leverages a fixed offline dataset to learn a policy that achieves high performance in online evaluation. Previous work has focused on policy conservativeness to prevent over-estimation and mitigate out-of-distribution actions during evaluation (Fujimoto et al., 2019; Wu et al., 2019b; Peng et al., 2019; Kostrikov et al., 2021; Kumar et al., 2020). Imitation Learning involves learning the behavior policy from the dataset, instead of using the reinforcement learning. Some studies in Imitation Learning also consider the potential multi-modality of the dataset (Li et al., 2017; Kuefler & Kochenderfer, 2017; Wang et al., 2017; Igl et al., 2023; Shafiullah et al., 2022; Wu et al., 2019a). Offline skill discovery shares similarities with learning diverse policies, but its goal is to improve performance in downstream tasks through the acquisition of skills. These approaches employ similar methods to model the latent variable and capture different skills (Laskin et al., 2022; Villecroze et al., 2022).

**Diversity in RL**   Diversity plays a crucial role in Reinforcement Learning algorithms. In some works, diversity is supposed to enhance the overall quality by means of encouraging exploration

(Hong et al., 2018) or better opponent modeling (Fu et al., 2022). Besides studies on diverse policies, skill discovery also shares the same requirement for diverse skills (Eysenbach et al., 2019; Campos et al., 2020; Sharma et al., 2020; Achiam et al., 2018). Other works consider optimizing both diversity and high quality in the optimization (Masood & Doshi-Velez, 2019; Zhang et al., 2019b; Zhou et al., 2022). It is important to mention that studies on diversity in the online environment promote exploration or skill discovery. However, in this paper, our focus is on diversity in the offline setting, which is beneficial for opponent modeling and various applications such as game AI and autonomous driving.

**Quality-Diversity Optimization** Some research in the field of evolutionary algorithms also focuses on discovering diverse policies. They formulate the problem as the Quality-Diversity Optimization problem (Pugh et al., 2016; Cully & Demiris, 2018), where "quality" refers to the policy's performance, and "diversity" emerges from the evolutionary iterations. An algorithm called MAP-Elites has been developed to generate diverse and high-quality policies (Cully, 2015; Mouret & Clune, 2015). Subsequent studies in this area aim to improve the efficiency of the evolutionary algorithm by combining it with policy gradient methods (Pierrot et al., 2022; Nilsson & Cully, 2021; Pierrot et al., 2022) or evolution strategies (Colas et al., 2020; Wang et al., 2022).

## 3 PRELIMINARY

In this paper, we consider a Markov Decision Process (MDP) defined by the tuple $(\mathcal{S}, \mathcal{A}, P, r, \rho_0, \gamma)$, where $\mathcal{S}$ is the state space, $\mathcal{A}$ is the action space, and $P : S \times A \times S \to \mathbb{R}$ is the transition function. The reward function is denoted by $r : \mathcal{S} \times \mathcal{A} \to \mathbb{R}$. We use $\rho_0 : \mathcal{S} \to \mathbb{R}$ to denote the initial state distribution, and $\gamma$ to denote the discount factor. In standard reinforcement learning (RL), a learning agent optimizes its polity $\pi : \mathcal{S} \times \mathcal{A} \to \mathbb{R}$ to maximize the expected cumulative discounted return $J(\pi) = \mathbb{E}_\pi \sum_{t=0}^{T} \gamma^t r(s_t, a_t)$, where $s_0 \sim \rho_0$, $a_t \sim \pi(a_t|s_t)$, and $s_{t+1} \sim P(s_{t+1}|s_t, a_t)$. The value function $V^\pi(s_t)$ corresponds to the expected return of policy $\pi$ at state $s_t$, and the action value function $Q^\pi(s_t, a_t)$ refers to the expected return obtained by playing action $a_t$ at state $s_t$ and then following $\pi$. The advantage function $A^\pi(s_t, a_t)$ is defined as $A^\pi(s_t, a_t) = Q^\pi(s_t, a_t) - V^\pi(s_t)$. We use $d_\pi(s) = \sum_{t=0}^{T} \gamma^t p(s_t = s|\pi)$ to denote the unnormalized discounted state distribution induced by policy $\pi$.

In offline RL, an agent learns from a pre-collected dataset $\mathcal{D}$ consisting of multiple trajectories without online interaction with the environment. This paper further assumes that the dataset $\mathcal{D}$ contains behaviors of heterogeneous policies $\{\beta^{(1)}, \beta^{(2)}, \ldots, \beta^{(K)}\}$. The assumption is not restrictive for many real-world scenarios. For instance, in online gaming and autonomous driving, the dataset usually contains behaviors of a number of different agents and humans, each of which may possess distinct behavioral patterns. The objective of our approach, termed Stylized Offline RL (SORL), is to learn a set of high-quality and diverse policies $\{\pi^{(1)}, \pi^{(2)}, \ldots, \pi^{(m)}\}$ from the most representative behaviors exhibited by the dataset $\mathcal{D}$.

## 4 METHOD

In order to leverage heterogeneous datasets and strike a balance between policy diversity and performance, we propose Stylized Offline RL (SORL), a novel two-step framework consisting of (1) EM-based style clustering and (2) advantage-weighted style learning. In style clustering, for each cluster that represents a distinct and dominant behavioral style, SORL assigns different weights to trajectories in the dataset. A trajectory's weight reflects its posterior probability of belonging to that style. Subsequently, SORL incorporates stylized advantage weighted regression to learn diverse and high-quality policies by constraining each policy to be conservative with respect to the corresponding weighted set of data. The SORL algorithm is illustrated in Algorithm 1.

### 4.1 EM-BASED STYLE CLUSTERING

The latent variable model is a natural way to disentangle a dataset generated by diverse policies (Li et al., 2017; Wang et al., 2017; Laskin et al., 2022). In our problem, each trajectory in the dataset is generated by an unknown latent policy. Therefore, drawing inspiration from the expectation-maximization (EM) algorithm, SORL alternates between trajectory clustering and policy learning to

extract the most representative diverse behaviors from the heterogeneous dataset. SORL learns a set of $m$ most representative policies $\{\mu^{(1)}, \mu^{(2)}, \ldots, \mu^{(m)}\}$ by maximizing the log-likelihood:

$$LL(\{\mu^{(i)}\}) = \sum_{\tau \in \mathcal{D}} \log \left[ \sum_{i=1}^{m} p(\tau|z=i)p(\mu^{(i)}) \right] \tag{1}$$

where $\tau$ denotes a trajectory in the dataset, $z$ is the latent variable indicating which policy $\tau$ belongs to, $p(\tau|z=i)$ is the probability of $\tau$ sampled under policy $\mu^{(i)}$, and $p(\mu^{(i)})$ is the prior distribution. The policies serve as the generative model for generating the dataset. The main challenge is to identify latent $z$. The EM-based algorithm iteratively updates the policies and the estimation $\hat{p}$ for the posteriors $p(z|\tau)$. Upon its convergence, the estimated posterior distribution $\hat{p}(z|\tau)$ offers a clustering of trajectories in the dataset, and the policy $\mu^{(i)}$ imitates the behavior in cluster $i$.

### 4.1.1 E-STEP

In the E-step, SORL calculates the estimated posterior distribution of the latent variable $\hat{p}(z|\tau)$ with respect to the current policies $\{\mu^{(i)}\}$. Formally, $\hat{p}(z=i|\tau) \propto p(\mu^{(i)}) \prod_{(s,a)\in\tau} \mu^{(i)}(a|s)$. In this paper, we assume a uniform prior. Empirically, we find that the long trajectory horizon leads to numerical instability when multiplying probabilities. Hence, we employ an alternative approach to estimate the latent variable. Since all steps in the trajectory $\tau$ share the same latent variable, the posterior distribution $\hat{p}(z|\tau)$ is estimated by averaging across all the samples in a trajectory.

$$\hat{p}(z=i|\tau) \approx \frac{1}{Z} \sum_{(s,a)\in\tau} \mu^{(i)}(a|s) \tag{2}$$

where $Z$ is a normalizing factor. SORL uses equation 2 to calculate the posteriors for all trajectories of the dataset in the E-step.

### 4.1.2 M-STEP

According to Jensen's inequality (Durrett, 2019),

$$LL(\{\mu^{(i)}\}) \geq \sum_{\tau \in \mathcal{D}} \sum_{i=1}^{m} \left[ \hat{p}(z=i|\tau) \log \frac{p(\tau|z=i)p(\mu^{(i)})}{\hat{p}(z=i|\tau)} \right]. \tag{3}$$

In the M-step, with the posteriors frozen, the policies $\{\mu^{(i)}\}_{i=1}^{m}$ are updated to maximize the right side of inequality 3, which is a lower bound of the log likelihood in equation 1. In order to maximize the objective $\sum_{\tau \in \mathcal{D}} \sum_{i=1}^{m} [\hat{p}(z=i|\tau) \log p(\tau|z=i)]$, SORL utilizes weighted imitation learning to minimize the following loss:

$$Loss(\{\mu^{(i)}\}) = \frac{1}{|\mathcal{D}|} \sum_{\tau \in \mathcal{D}} \sum_{i=1}^{m} \hat{p}(z=i|\tau) \sum_{(s,a)\in\tau} \log \mu^{(i)}(a|s) \tag{4}$$

The style clustering algorithm shares the same convergence result with the original EM algorithm, and it is guaranteed to converge to a saddle point (Ng et al., 2012).

### 4.2 ADVANTAGE-WEIGHTED STYLE LEARNING

In the second step, SORL further improves the policies' performances with stylized advantage-weighted regression. Formally, SORL learns a set of policies $\{\pi^{(1)}, \pi^{(2)}, \ldots, \pi^{(m)}\}$ through the following constrained optimization problem:

$$\forall i \in [m], \pi^{(i)} = \arg\max J(\pi^{(i)})$$

$$s.t. \; \mathbb{E}_{s \sim d_{\mu^{(i)}}(s)} D_{KL}(\pi^{(i)}(\cdot|s) || \mu^{(i)}(\cdot|s)) \leq \epsilon, \quad \int_a \pi^{(i)}(a|s)da = 1, \; \forall s. \tag{5}$$

The policies learn to maximize cumulative return to avoid degenerate behavior, while still remaining diverse through constraining the KL divergence between $\mu^{(i)}$ and $\pi^{(i)}$ to be small. Maximizing the cumulative return $J(\pi^{(i)})$ is equivalent to maximizing the expected improvement $\eta(\pi^{(i)}) = J(\pi^{(i)}) - J(\mu^{(i)})$ $(i = 1 \ldots m)$. According to previous works, the expected improvement can be expressed in terms of advantage: (Schulman et al., 2015)

$$\eta(\pi^{(i)}) = \mathbb{E}_{s \sim d_{\pi^{(i)}}(s)} \mathbb{E}_{a \sim \pi^{(i)}(a|s)} [A^{\mu^{(i)}}(s,a)] \tag{6}$$

Equation 6 poses challenges in optimization due to the unknown discounted state distribution $d_{\pi^{(i)}}$. To address this, a common approach is to substitute $d_{\pi^{(i)}}$ with $d_{\mu^{(i)}}$ to provide a good estimate of $\eta(\pi^{(i)})$ (Schulman et al., 2015; Peng et al., 2019). It has been proved that the error can be bounded by the KL divergence between $\pi^{(i)}$ and $\mu^{(i)}$ (Schulman et al., 2015), which has already been constrained in our objective 5. Therefore, we optimize the following objective 11.

$$\forall i \in [m],\, \pi^{(i)} = \arg\max \mathbb{E}_{s \sim d_{\mu^{(i)}}(s)} \mathbb{E}_{a \sim \pi^{(i)}(\cdot|s)} A^{\mu^{(i)}}(s, a)$$

$$s.t.\ \mathbb{E}_{s \sim d_{\mu^{(i)}}(s)} D_{KL}(\pi^{(i)}(\cdot|s)||\mu^{(i)}(\cdot|s)) \leq \epsilon, \quad \int_a \pi^{(i)}(a|s)da = 1,\ \forall s. \tag{7}$$

In advantage-weighted style learning, we approximate $A^{\mu^{(i)}}(s, a)$ by $A^{\mu}(s, a)$, where $\mu$ represents the policy distribution of the entire dataset. This approximation is made because $A^{\mu}(s, a)$ often has higher quality than $A^{\mu^{(i)}}$. Subsequently, we calculate the Lagrangian of the optimization problem:

$$\mathbb{E}_{s \sim d_{\mu^{(i)}}(s)} \left[ \mathbb{E}_{a \sim \pi^{(i)}(\cdot|s)} A^{\mu}(s, a) + \lambda(\epsilon - D_{KL}(\pi^{(i)}(\cdot|s)||\mu^{(i)}(\cdot|s))) \right] + \int_s \alpha_s (1 - \int_a \pi^{(i)}(a|s)da) \tag{8}$$

Note that the policy $\pi^{(i)}$ of each style can be optimized independently. Taking derivative with respect to $\pi_i(a|s)$, we can obtain the closed-form solution $\pi^{(i)*}(a|s) \propto \mu^{(i)}(a|s)exp(\frac{1}{\lambda}A^{\mu}(s, a))$. Finally, we project $\pi^{(i)*}$ to $\pi_\theta^{(i)}$ paramterized by $\theta$ by minimizing the KL divergence between them. It can be proved that the parameterized policy of $i$-th style $\pi_\theta^{(i)}$ can be learned by minimizing the following loss. Please refer to Appendix D for detailed proofs.

$$Loss(\pi_\theta^{(i)}) = -\mathbb{E}_{\tau \sim \mathcal{D}} \hat{p}(z = i|\tau) \sum_{(s,a) \in \tau} \log \pi_\theta^{(i)}(a|s) \exp\left( \frac{1}{\lambda} A^{\mu}(s, a) \right). \tag{9}$$

Compared with the previous offline RL algorithm AWR (Peng et al., 2019) that learns a single policy, SORL learns a set of diverse policies by assigning different weights to trajectories in the dataset. SORL leverages the estimated posterior distribution calculated in step one to force different policies to focus on different clusters of trajectories that possess distinct behavior patterns.

The two-step SORL algorithm is illustrated in Algorithm 1. SORL leverages the EM-based style clustering to extract distinct styles from the dataset. Furthermore, it takes into account both diversity and quality improvements to perform advantage-weighted learning. As a result, the SORL algorithm extracts from the heterogeneous dataset a collection of diverse policies with high quality.

## 5 EXPERIMENTS

In this experimental section, we aim to address the following questions: (1) Can SORL derive high-performing policies that exhibit diverse behaviors from an offline heterogeneous dataset originating from various sources? (2) How does SORL compare to prior methods that focus on learning diverse behaviors from an offline heterogeneous dataset? (3) Is SORL suitable for complex, real-world tasks, especially those involving large-scale datasets collected from human users?

Our experiments encompass a range of tasks and offline datasets. For each experiment, we train a set of policies and evaluate them based on three criteria: *quality*, *diversity*, and *consistency*. These criteria respectively reflect the performance, behavioral diversity, and whether the learned diversity is inherent in the offline dataset, addressing the question (1). To answer question (2), we compare SORL against two prior methods: Off-RLPMM (Wu et al., 2019a) and InfoGAIL (Li et al., 2017). This comparison spans three sets of experiments, each with increasing levels of difficulty.

The first experiment involves a grid-shooting game where the dataset is manually collected from two distinct winning strategies. This experiment allows for a detailed analysis of the learned policies as we have full control over the game environment and access to the ground truth styles in the dataset. The second experiment features a set of Atari games. The datasets for these games are recorded from human players in a controlled, semi-frame-by-frame manner (Zhang et al., 2019a). As a result, the quality of the dataset is relatively high. This experiment assesses the performance of the methods when learning from heterogeneous datasets with minimal interference from inconsistent data quality. The final experiment centers on the popular basketball video game "Dunk City Dynasty" (FuxiRL, 2023). Here, datasets are recorded directly from online players, leading to greater diversity and

---

**Algorithm 1** Stylized Offline RL (SORL)

---

1: Input: offline heterogeneous dataset $\mathcal{D}$, number of policies $m$
2: Output: learned diverse policies $\pi_\theta^{(1)} \cdots \pi_\theta^{(m)}$
3: Initialize policies $\pi_\theta^{(1)} \cdots \pi_\theta^{(m)}$ and $\mu^{(1)} \cdots \mu^{(m)}$
4: # EM-based style clustering
5: **while** not converged **do**
6:    # E-step
7:    Calculate the posterior probability of styles $\hat{p}(z|\tau)$ with Equation 2
8:    # M-step
9:    Calculate the loss function $Loss(\{\mu^{(i)}\})$ defined in Equation 4
10:    Update $\{\mu^{(i)}\}$ by taking one step of gradient descent with respect to $Loss(\{\mu^{(i)}\})$
11: **end while**
12: # Advantage-weighted style learning
13: Calculate the posterior probability of styles $\hat{p}(z|\tau)$ with Equation 2
14: **while** not converged **do**
15:    Calculate the loss function $Loss(\{\pi_\theta^{(i)}\})$ defined in Equation 9
16:    Update $\{\pi_\theta^{(i)}\}$ by taking one step of gradient descent with respect to $Loss(\{\pi_\theta^{(i)}\})$
17: **end while**

---

considerable variations in data quality. This experiment evaluates the effectiveness of the methods in handling complex, real-world tasks using datasets collected from a diverse group of humans, addressing the question (3).

In the following, we introduce the baseline methods and the evaluation criteria, and then present the results of the three experiments.

## 5.1 BASELINES

**Off-RLPMM**: RLPMM (Wu et al., 2019a) adopts a similar EM framework, but it is not directly applicable in our offline setting because it requires additional online interaction to train a classifier for labeling all the trajectories in the M-step. We implement an offline version of RLPMM (Off-RLPMM) that adopts a similar M-step as SORL, except that Off-RLPMM partitions the dataset into disjoint subsets instead of using SORL's weighting scheme.

**InfoGAIL**: InfoGAIL (Li et al., 2017) studies imitation learning from multimodal datasets. Different from the EM stage of SORL, InfoGAIL infers the latent style of trajectories by maximizing the mutual information between the latent codes and trajectories.

In both baselines, we employ Behavior Cloning (BC) as the imitation learning algorithm. Additionally, in contrast to the two-stage algorithm SORL, both baselines do not have a second policy improvement stage. Detailed explanations are provided in Appendix E.

## 5.2 EVALUATION CRITERIA

We evaluate the learned policy with the following three criteria.

**Quality**: The quality of a policy is assessed by the episode return of trajectories during evaluation. It reflects the performance of the policy in completing tasks.

**Diversity**: Diversity is determined by the entropy of styles identified from the learned policies. With learned clustering $\hat{p}$, each trajectory is classified into a style with the highest probability. Diversity is then calculated as the entropy of the distribution of styles over the whole dataset. In conjunction with the later-introduced consistency metric, a higher entropy suggests that policies exhibit diverse, distinguishable behaviors.

$$p_{polularity}(z = i) = \frac{1}{|\mathcal{D}|} \sum_{\tau \in \mathcal{D}} \mathbb{I}(\arg\max_j \{\hat{p}(z = j|\tau)\} = i)$$

$$Diversity := Entropy(p_{popularity}(\cdot)) \tag{10}$$

**Consistency**: The consistency measures the likelihood that the learned policy generates actions corresponding to the specific cluster it is associated with within the dataset. Evaluating consistency is crucial to ensure that the behaviors we have learned are inherent in the heterogeneous datasets, rather than being unknown out-of-distribution actions, which could also artificially inflate diversity scores.

### 5.3 DIDACTIC EXAMPLE: GRID SHOOTING

In the grid shooting game, a player navigates in a 9x9 grid world, controlling a character that engages in combat against an AI opponent. Both the player and the AI have the ability to move in four directions within the grid, shoot at each other, and gather stars that randomly appear in one of the grid cells. Rewards and termination conditions are distributed based on the subsequent events:

- **Shooting:** If the player and the AI opponent are positioned in the same row or column, they have the opportunity to shoot at each other. A successful shot ends the game, with the shooter being declared the winner and receiving a reward of 10.

- **Star Collection:** A star is randomly placed in a grid cell, and its location is known to both the player and the AI opponent. Whichever party reaches the location of the star earns a reward of 1. After the star is collected, it reappears randomly in another grid cell. If no successful shooting occurs, the game terminates at the 100th step, and the participant with the higher accumulated rewards is announced as the winner.

| Methods | Learned policies | Reward (shoot) | Reward (star) | Winning rate |
|---------|------------------|----------------|---------------|--------------|
| SORL | policy 1 | $2.6 \pm 0.1$ | $2.2 \pm 1.3$ | $51.0 \pm 0.1\%$ |
|  | policy 2 | $1.1 \pm 0.4$ | $6.3 \pm 0.2$ | $59.3 \pm 0.0\%$ |
| Off-RLPMM | policy 1 | $2.7 \pm 0.1$ | $0.0 \pm 0.0$ | $26.5 \pm 0.0\%$ |
|  | policy 2 | $0.0 \pm 0.0$ | $8.6 \pm 0.1$ | $54.4 \pm 0.0\%$ |
| InfoGAIL | policy 1 | $1.7 \pm 0.1$ | $4.9 \pm 0.7$ | $52.0 \pm 5.7\%$ |
|  | policy 2 | $1.5 \pm 0.3$ | $3.5 \pm 0.4$ | $58.7 \pm 7.4\%$ |

Table 1: The reward distribution and the winning rate of the two policies learned from the heterogeneous dataset.

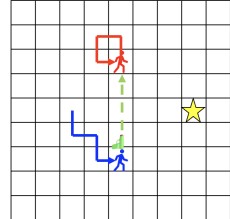
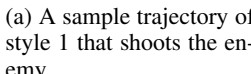
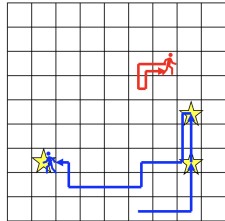

(a) A sample trajectory of style 1 that shoots the enemy.

(b) A sample trajectory of style 2 that gets the stars.

Figure 1: Visualization of policies learned by SORL, capturing the main playing styles in the dataset.

In the Grid Shooting environment, there are two distinct winning strategies or playing styles. The first strategy emphasizes shooting the enemy, while the second focuses on collecting stars and avoiding direct confrontations with the enemy, as depicted in Figure 1. We constructed our dataset by combining trajectories from both playing styles. Using this dataset, we trained two policies with SORL and compared them with two baseline methods. The rewards earned for each event and the final winning rates are detailed in Table 1.

From the table, it is evident that SORL successfully captures the core difference between the two playing styles. This is reflected in the distinct reward distributions for shooting and star collecting across the two learned policies. Furthermore, the winning rates of the policies are promising, underscoring their robust performance. In contrast, while Off-RLPMM also captures the two playing styles, the winning rates of its policies are lower than those of SORL. InfoGAIL, on the other hand,

yields a competitive winning rate but fails to differentiate between the styles of the policies. Figure 1 shows example trajectories for the two learned policies of SORL. More detailed quantitative results including the quality, diversity and consistency is presented in Appendix A.2.

## 5.4 ATARI GAMES

In this experiment, we focus on six Atari games, including SpaceInvaders, MsPacman, MontezumaRevenge, Enduro, Riverraid, and Frostbite. For each game, We train three policies using the dataset provided by Atari Head (Zhang et al., 2019a), a large-scale dataset of human players. The dataset is recorded in a controllable semi-frame-by-frame manner, ensuring high data quality. The experimental results, which include metrics on quality, diversity, and consistency, are presented in Table 2. A visual representation of these results, using a radar plot, is available in Figure 2.

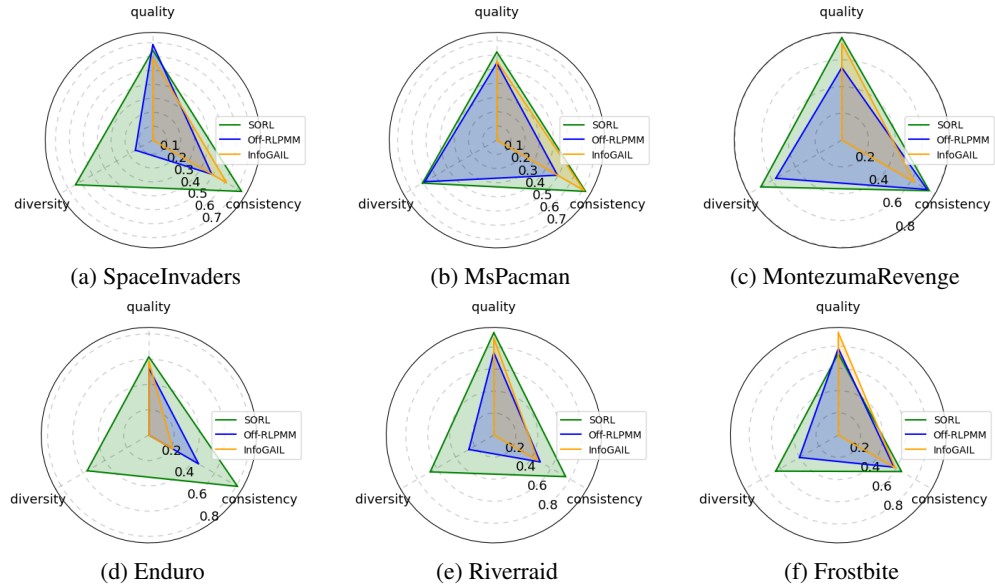

Figure 2: The radar plot of quality, diversity, and consistency in the Atari environment.

| Games | Methods | Quality | Diversity | Consistency |
|---|---|---|---|---|
| SpaceInvaders | SORL | $387.1 \pm 33.3$ | $0.96 \pm 0.11$ | $94.7 \pm 0.1\%$ |
| | Off-RLPMM | $412.5 \pm 16.3$ | $0.22 \pm 0.15$ | $89.8 \pm 0.2\%$ |
| | InfoGAIL | $353.2 \pm 19.3$ | $0.0 \pm 0.0$ | $92.3 \pm 0.7\%$ |
| MsPacman | SORL | $622.2 \pm 65.3$ | $0.91 \pm 0.09$ | $94.5 \pm 0.2\%$ |
| | Off-RLPMM | $543.4 \pm 80.2$ | $0.88 \pm 0.09$ | $90.0 \pm 0.1\%$ |
| | InfoGAIL | $558.0 \pm 137.8$ | $0.00 \pm 0.00$ | $94.2 \pm 0.1$ |
| MontezumaRevenge | SORL | $306.7 \pm 19.1$ | $1.05 \pm 0.03$ | $95.1 \pm 0.1\%$ |
| | Off-RLPMM | $216.7 \pm 95.3$ | $0.86 \pm 0.13$ | $94.8 \pm 0.1$ |
| | InfoGAIL | $290.0 \pm 58.6$ | $0.00 \pm 0.00$ | $92.3 \pm 1.1$ |
| Enduro | SORL | $371.1 \pm 2.9$ | $0.84 \pm 0.25$ | $96.2 \pm 0.1\%$ |
| | Off-RLPMM | $317.2 \pm 7.5$ | $0.00 \pm 0.00$ | $89.1 \pm 0.6\%$ |
| | InfoGAIL | $351.3 \pm 31.0$ | $0.00 \pm 0.00$ | $84.4 \pm 0.8\%$ |
| Riverraid | SORL | $1931.2 \pm 432.1$ | $1.00 \pm 0.04$ | $95.1 \pm 0.2\%$ |
| | Off-RLPMM | $1748.6 \pm 113.7$ | $0.39 \pm 0.29$ | $89.7 \pm 3.9\%$ |
| | InfoGAIL | $1882.6 \pm 149.8$ | $0.00 \pm 0.00$ | $89.2 \pm 1.5\%$ |
| Frostbite | SORL | $2056.0 \pm 391.9$ | $0.97 \pm 0.04$ | $93.5 \pm 0.4\%$ |
| | Off-RLPMM | $2183.2 \pm 502.4$ | $0.60 \pm 0.07$ | $91.6 \pm 0.0\%$ |
| | InfoGAIL | $2584.7 \pm 206.5$ | $0.00 \pm 0.00$ | $91.8 \pm 0.3\%$ |

Table 2: The experiment results of quality, diversity and consistency in the Atari environment.

From Table 2, we see results consistent with those in the grid shooting game. SORL outperforms the baseline methods, attaining the highest scores in both diversity and consistency for all games, and

secures the top quality score in four out of the six games. While Off-RLPMM does exhibit some diversity, its policy quality is weaker. Conversely, InfoGAIL achieves competitive quality scores but struggles to learn diverse policies. Visualizations of the stylized policies are in Appendix H.

## 5.5 VIDEO GAME APPLICATION

Dunk City Dynasty" (FuxiRL, 2023) is an online mobile game where players control a character to play in a 3v3 basketball match. The game presents a formidable challenge to existing algorithms due to its high-dimensional state and action spaces. For this experiment, we collect a dataset of over 100,000 steps, directly from online players. Compared to the other two experiments, the dataset used here has notable variations in both behavioral styles and data quality. Same as Atari games, we train three policies with SORL and compare them with two baseline methods. The experimental results on quality, diversity, and consistency are presented in Table 3. The results highlight that SORL consistently outperforms in all three evaluation metrics, underscoring its robustness and adaptability in handling complex, real-world tasks, especially when working with large-scale datasets from diverse human players.

| Methods | Quality | Diversity | Consistency |
|---------|---------|-----------|-------------|
| SORL | $5.3 \pm 3.4$ | $1.05 \pm 0.0$ | $40.7 \pm 4.6\%$ |
| Off-RLPMM | $1.7 \pm 0.5$ | $0.51 \pm 0.18$ | $38.6 \pm 5.4\%$ |
| InfoGAIL | $5.0 \pm 0.8$ | $0.73 \pm 0.22$ | $39.8 \pm 3.3\%$ |

Table 3: The experiment results of quality, diversity and consistency in the "Dunk City Dynasty" environment.

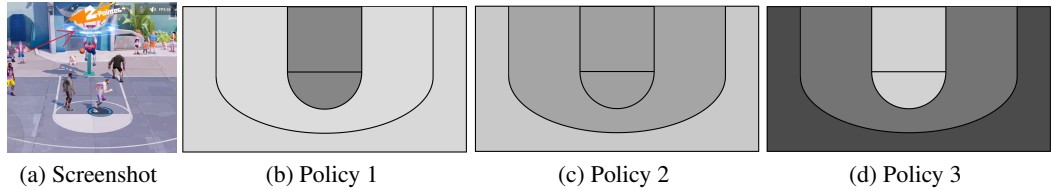

| (a) Screenshot | (b) Policy 1 | (c) Policy 2 | (d) Policy 3 |

Figure 3: A scene in the game, and the shooting areas favored by each policy. The grayscale represents the distribution of shooting action locations, with darker shades indicating higher probabilities and lighter shades signifying lower probabilities.

We observed distinct shooting range preferences among the three policies. For example, policy 1 tends to favor short-range shots, while policy 3 is inclined towards long-range shots. The preferred shooting areas for each policy are visualized in Figure 3. Additional screenshots showcasing typical behaviors of these policies can be found in Figure 5 in the Appendix B. To better illustrate the behaviors exhibited by the learned policies, we have included gameplay videos in the supplementary materials..

## 6 CONCLUSION

In this paper, we explored the extraction of diverse and high-quality behaviors from offline heterogeneous datasets. These datasets, sourced from multiple origins, inherently possess a multimodal data distribution with inconsistent data quality. Such a setting poses significant challenges to existing methods, which can be affected by low-quality data or lack the control to differentiate the behavioral while learning. To address these challenges, we introduced the Stylized Offline Reinforcement Learning (SORL) framework. SORL employs an EM-based style clustering combined with advantage-weighted policy learning. This design not only optimizes the performance of the policies but also preserves the inherent behavioral diversity found in heterogeneous datasets. Through extensive experiments, we compared SORL with two prior methods across various tasks and heterogeneous offline datasets. Our results underscore SORL's superior capability in extracting behaviors that are both high-quality and diverse. Our future work aims to offer a more flexible formulation of policy behavior, potentially allowing shared behaviors between policies, making it more relevant for real-world applications. Furthermore, we plan to integrate adaptive task learning using the diverse policies derived from SORL, enabling dynamic switching in different task scenarios.

## 7    REPRODUCIBILITY STATEMENT

The source code for the SORL algorithm and the baselines used in this study are included in the supplementary materials. The proof sketch can be found in Section 4.2, and a comprehensive proof is provided in the Appendix 4.2. For the Grid Shooting environment, the data processing steps involve using the state and action directly from the environment output. As for the Atari Head dataset, the processing steps follow the methodology outlined in the original package.

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

**Appendix**

## A GRID SHOOTING

### A.1 ENVIRONMENT DETAILS

The state space dimension is 56. The action space is a discrete action space with 9 actions. The opponent is an agent with a fixed strategy that move randomly, and shoot with a probability once the shooting action cools down.

### A.2 QUANTITATIVE EXPERIMENT RESULTS OF THE GRID SHOOTING ENVIRONMENT

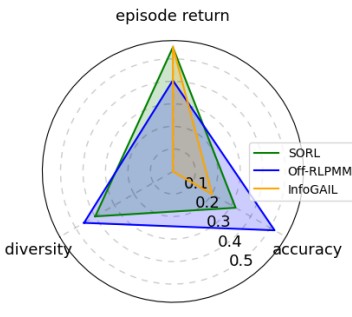

Figure 4: The radar plot of quality, diversity and consistency in the Grid Shooting environment.

|  | Quality | Diversity | Consistency |
|---|---|---|---|
| SORL | $55.1 \pm 4.6\%$ | $0.60 \pm 0.07$ | $86.4 \pm 0.4\%$ |
| Off-RLPMM | $40.4 \pm 0.7\%$ | $0.68 \pm 0.01$ | $90.5 \pm 0.0\%$ |
| InfoGAIL | $55.3 \pm 2.1\%$ | $0.00 \pm 0.00$ | $84.0 \pm 0.3\%$ |

Table 4: The experiment results of quality, diversity and consistency in the Grid Shooting environment.

## B "DUNK CITY DYNASTY"

### B.1 ENVIRONMENT DETAILS

The state space dimension is 468, including global state, allies' states and enemies' states. The action space is a discrete space with 52 actions. The opponent in evaluation is an agent with moderate strength, that is learned by vanilla Behavioral Cloning for similar training steps (100,000 steps).

### B.2 SCREENSHOTS IN "DUNK CITY DYNASTY"

Table 3 presents three metrics in the experiment, illustrating SORL's ability to learn diverse policies while achieving satisfactory performance. Figure 5 shows screenshots from the videos showcasing each style's self-playing behavior. Supplementary materials include videos that provide additional visual demonstrations. For instance, style 1 demonstrates a preference for shooting in the short range, while style 3 favors long-range shots. The videos reveal that the proportion of goals scored inside the restricted area is $83.3\%$, $66.7\%$, and $33.3\%$ for styles 1, 2 and 3, respectively. Furthermore, the proportion of two-point shots is $91.6\%$, $88.9\%$ and $66.7\%$. Figure 4 depicts the probabilities of shooting in different regions. In Figure 6, (a) and (c) showcase typical shots of Style 1 and 3. Panel (a) demonstrates shots within the restricted zone, while panel (c) displays long- range shots. Style 2 is characterized by ball possession and running across the arena. The videos indicate that Style 2 has the fewest number of passes (41, 29, 39, respectively), and the lowest proportion of goals scored directly after a pass ($27.7\%$, $22.2\%$, $44.4\%$).

### B.3 THE PLOT OF SHOOTING POSITIONS

We plot the shooting positions in Figure 6 based on the gameplay data. The shooting positions in the plot corresponds to the distribution in Figure 3.

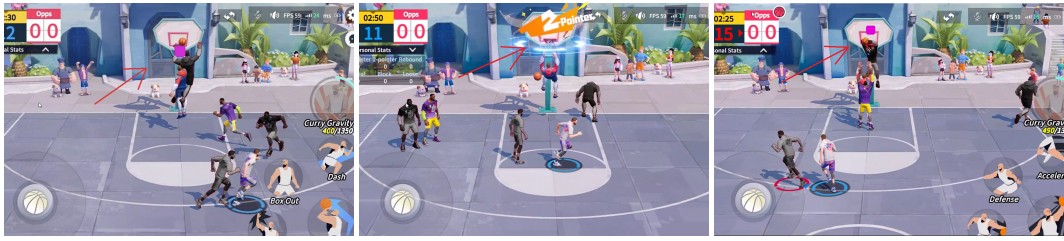

(a) Style 1 prefers short-range shots.

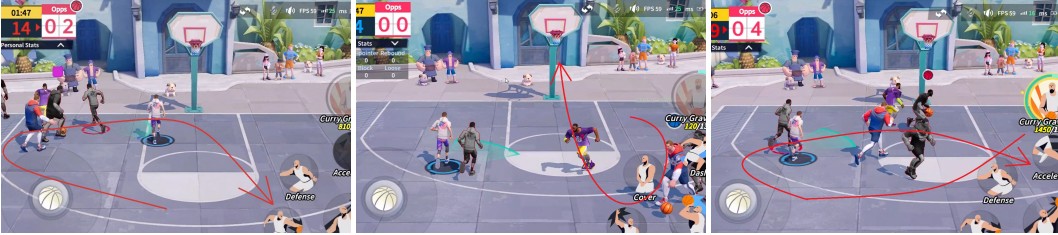

(b) Style 2 prefers ball handling run.

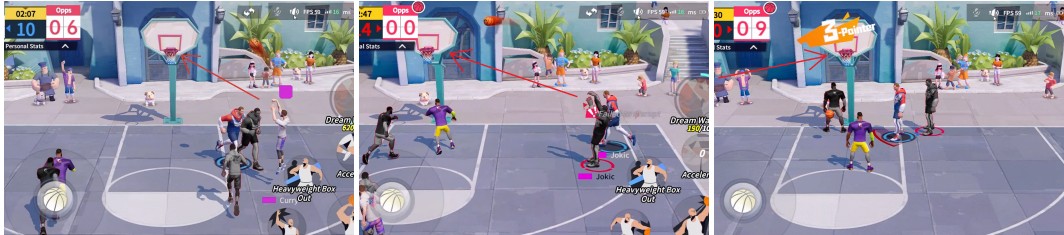

(c) Style 3 prefers long-range shots.

Figure 5: The screenshots of 3 styles.

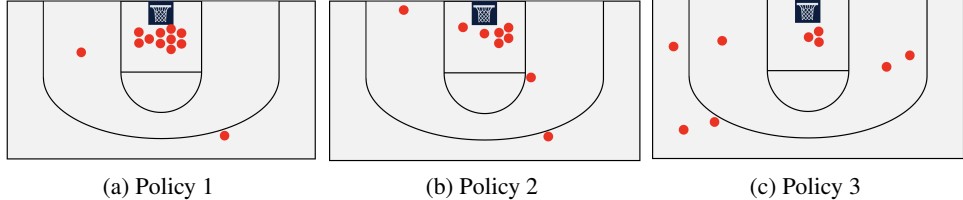

(a) Policy 1        (b) Policy 2        (c) Policy 3

Figure 6: The Plot of Shooting Positions.

## C  ABLATION STUDY

In this section, we performed ablation experiments on the advantage-weighted policy learning. Based on the algorithm description provided in Section 4, the SORL algorithm can be divided into two parts. Therefore, when the advantage-weighted policy learning is excluded, SORL reduces to the EM-based style clustering. Table 5 presents the results, indicating that the consistency and diversity metrics show similar performance, while the quality metric improves as a result of the advantage-weighted style learning.

|  |  | Quality | Diversity | Consistency |
|---|---|---|---|---|
| SpaceInvaders | SORL | $387.1 \pm 33.3$ | $0.96 \pm 0.11$ | $94.7 \pm 0.1\%$ |
|  | SORL w/o advantage | $349.9 \pm 50.2$ | $0.90 \pm 0.14$ | $93.9 \pm 0.2\%$ |
| MsPacman | SORL | $622.2 \pm 65.3$ | $0.91 \pm 0.09$ | $94.5 \pm 0.2\%$ |
|  | SORL w/o advantage | $590.0 \pm 40.0$ | $1.02 \pm 0.05$ | $93.9 \pm 0.2\%$ |
| MontezumaRevenge | SORL | $306.7 \pm 19.1$ | $1.05 \pm 0.03$ | $95.1 \pm 0.1\%$ |
|  | SORL w/o advantage | $211.1 \pm 39.7$ | $1.01 \pm 0.05$ | $94.7 \pm 0.1\%$ |
| Enduro | SORL | $371.1 \pm 2.9$ | $0.84 \pm 0.25$ | $96.2 \pm 0.1\%$ |
|  | SORL w/o advantage | $405.2 \pm 9.7$ | $0.91 \pm 0.21$ | $95.8 \pm 0.1\%$ |
| Riverraid | SORL | $1931.2 \pm 432.1$ | $1.00 \pm 0.04$ | $95.1 \pm 0.2\%$ |
|  | SORL w/o advantage | $1842.4 \pm 282.2$ | $0.95 \pm 0.00$ | $94.1 \pm 0.2\%$ |
| Frostbite | SORL | $2056.0 \pm 391.9$ | $0.97 \pm 0.04$ | $93.5 \pm 0.4\%$ |
|  | SORL w/o advantage | $1787.8 \pm 250.6$ | $0.90 \pm 0.07$ | $92.4 \pm 0.1\%$ |
| DunkCityDynasty | SORL | $5.3 \pm 3.4$ | $1.05 \pm 0.0$ | $40.7 \pm 4.6\%$ |
|  | SORL w/o advantage | $0.7 \pm 0.9$ | $0.92 \pm 0.11$ | $39.8 \pm 5.7\%$ |

Table 5: The ablation study of SORL, comparing the SORL algorithm without advantage-weighted style learning.

# D  PROOF DETAILS

We provide detailed proof of solving the constrained optimization problem in Section 4.2. The original problem is as follows.

$$\forall i \in [m], \ \pi^{(i)} = \arg\max \mathbb{E}_{s \sim d_{\mu^{(i)}}(s)} \mathbb{E}_{a \sim \pi^{(i)}(\cdot|s)} A^{\mu^{(i)}}(s, a)$$

$$s.t. \ \mathbb{E}_{s \sim d_{\mu^{(i)}}(s)} D_{KL}(\pi^{(i)}(\cdot|s) || \mu^{(i)}(\cdot|s)) \le \epsilon, \tag{11}$$

$$\int_a \pi^{(i)}(a|s) da = 1, \ \forall s.$$

In advantage-weighted style learning, we approximate $A^{\mu^{(i)}}(s, a)$ by $A^\mu(s, a)$, where $\mu$ represents the policy distribution of the entire dataset. This approximation is made because $A^\mu(s, a)$ often has higher quality than $A^{\mu^{(i)}}$. Subsequently, we calculate the Lagrangian of the optimization problem:

$$L(\pi^{(i)}, \lambda, \alpha) = \mathbb{E}_{s \sim d_{\mu^{(i)}}(s)} \Big[ \mathbb{E}_{a \sim \pi^{(i)}(\cdot|s)} A^\mu(s, a)$$

$$+ \lambda(\epsilon - D_{KL}(\pi^{(i)}(\cdot|s) || \mu^{(i)}(\cdot|s)))) \Big] \tag{12}$$

$$+ \int_s \alpha_s (1 - \int_a \pi^{(i)}(a|s) da)$$

Differentiating on $\pi^{(i)}$,

$$\frac{\partial L}{\partial \pi^{(i)}(a|s)} = d_{\mu^{(i)}(s)} [A^\mu(s, a) - \lambda \log \pi^{(i)}(a|s) + \lambda \log \mu^{(i)}(a|s) - \lambda] - \alpha_s \tag{13}$$

We set $\frac{\partial L}{\pi^{(i)}(a|s)}$ to 0, and can get the closed-form solution $\pi^{(i)*}(a|s) = \frac{1}{Z^{(i)}(s)} \mu^{(i)}(a|s) \exp(\frac{1}{\lambda} A^\mu(s, a))$, where the normalization term $Z^{(i)}(s) = \exp(\frac{1}{d_{\mu^{(i)}(s)}} \frac{\alpha_s}{\lambda} + 1)$.

Finally, we project $\pi^{(i)*}$ to $\pi_\theta^{(i)}$ paramterized by $\theta$ by minimizing the KL divergence between them. $\mathcal{D}^{(i)}$ denotes the trajectories corresponding to style $i$ in the dataset, which is unknown. Hence, we

rewrite the expression by incorporating the probability of each trajectory being the $i$-th style.

$$\arg\min_{\theta} \mathbb{E}_{s\sim\mathcal{D}^{(i)}}[D_{KL}(\pi^{(i)*}(\cdot|s)||\pi_{\theta}^{(i)}(\cdot|s)]$$

$$= \arg\min_{\theta} \mathbb{E}_{s\sim\mathcal{D}^{(i)}}\Big[\int_a (\pi^{(i)*}(a|s)\log\pi^{(i)*}(a|s) - \pi^{(i)*}(a|s)\log\pi_{\theta}^{(i)}(a|s))\Big]$$

$$= \arg\min_{\theta} \mathbb{E}_{s\sim\mathcal{D}^{(i)}}\Big[\int_a (-\pi^{(i)*}(a|s)\log\pi_{\theta}^{(i)}(a|s))\Big]$$

$$= \arg\min_{\theta} \mathbb{E}_{s\sim\mathcal{D}^{(i)}}\Big[\int_a (-\frac{1}{Z^{(i)}(s)}\mu^{(i)}(a|s)\exp(\frac{1}{\lambda}A^{\mu}(s,a))\log\pi_{\theta}^{(i)}(a|s))\Big] \qquad (14)$$

$$= \arg\min_{\theta} -\mathbb{E}_{s\sim\mathcal{D}^{(i)}}\mathbb{E}_{a\sim\mathcal{D}^{(i)}}\log\pi_{\theta}^{(i)}(a|s)\frac{1}{Z^{(i)}(s)}\exp(\frac{1}{\lambda}A^{\mu}(s,a))$$

$$= \arg\min_{\theta} -\mathbb{E}_{\tau\sim\mathcal{D}^{(i)}}\log\pi_{\theta}^{(i)}(a|s)\frac{1}{Z^{(i)}(s)}\exp(\frac{1}{\lambda}A^{\mu}(s,a))$$

$$= \arg\min_{\theta} -\mathbb{E}_{\tau\sim\mathcal{D}}\hat{p}(z=i|\tau)\log\pi_{\theta}^{(i)}(a|s)\frac{1}{Z^{(i)}(s)}\exp(\frac{1}{\lambda}A^{\mu}(s,a))$$

Similar to AWR (Peng et al., 2019) and other prior work (Neumann & Peters, 2008; Siegel et al., 2020; Wang et al., 2018), we neglect the per-state normalizing constant $Z^{(i)}(s)$. The policy update can be expressed as follows:

$$\arg\min_{\theta} -\mathbb{E}_{\tau\sim\mathcal{D}}\hat{p}(z=i|\tau)\log\pi_{\theta}^{(i)}(a|s)\exp(\frac{1}{\lambda}A^{\mu}(s,a)) \qquad (15)$$

The original problem 11 has solution because it satisfies the Linear Indepence Constraint Qualification (LICQ).

## E   EXPERIMENT DETAILS

**Network structure**   We construct the network based on the default network of the relative task according to the codebase we use.  The network of Grid Shooting and Dunk City Dynasty is a 3-layer MLP, and the network of Atari environments has three convolution layers and two linear layers.

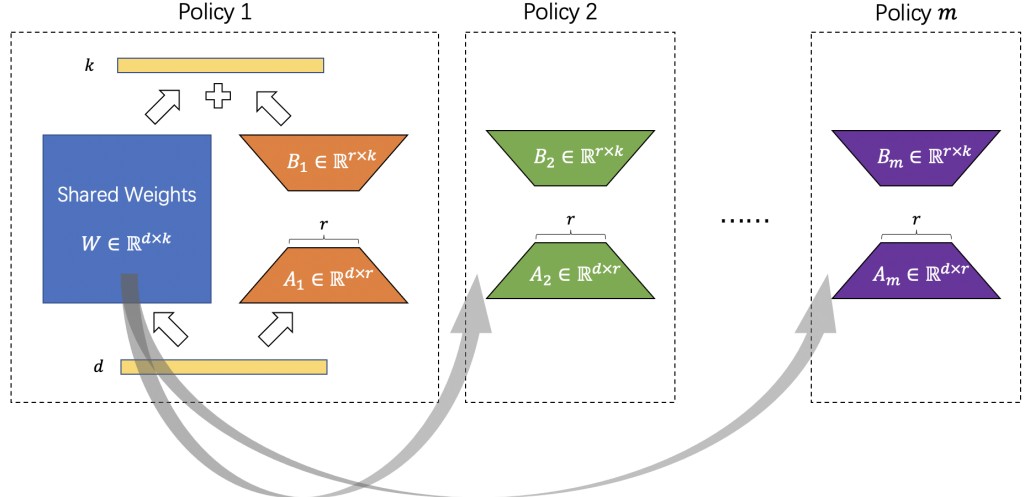

Figure 7: The usage of LoRA in the network structure of multiple policies.

Besides, in order to ensure balanced learning among all the styles, we share the main network and use a LoRA module to discriminate different styles. LoRA (Hu et al., 2021) is a widely used network structure, that substitutes the original matrix of the linear layer by a matrix $W$ and a low-

rank multiplication of two other matrices $A_i$ and $B_i$. In our setting as shown in Figure 7, the matrix $W$ is shared among all styles of policies, while $A_i, B_i$ varies. The input dimension $d$ and output dimension $k$ depends on the linear layer, and the rank $r$ is set to 9 in this work.

**Number of styles**   In Grid Shooting, we use 2 styles because we want to recover the two real styles of behaviors in the dataset. In other environments, we learn 3 styles from the dataset.

**Hyperparameters and other details**   In Grid Shooting, we use the batch size 128, offline dataset size 20000 and the number of epochs 10. In Atari environments, we use the batch size of $32 \times 5$ where we sample 32 $(s, a)$ pairs for 5 times, with the number of epochs 30. In Dunk City Dynasty, we use the batch size of around 300 and $100,000$ steps for the quality, diversity, and consistency metrics. We use $900,000$ steps for generating the playing videos of different styles.

We re-implement the InfoGAIL with continuous latent variables in all the environments. The 3 styles are extracted by applying $(1, 0, 0), (0, 1, 0), (0, 0, 1)$ as the latent variable to the policy model and get the 3 different policies.

The Grid Shooting and Atari results in the tables show the mean and standard deviation among 3 random seeds.

## F   ADDITIONAL METRICS EVALUATING DIVERSITY

In the main text, we use the diversity metric of popularity, which is a dataset-related metric that evaluates diversity. However, the definition of diversity, especially in the offline setting, is not unique. We provide additional diversity metrics in this part. The definitions of metrics are as follows.

1. **Skill metric** measures the dissimilarity of the skill set. The skill set is defined as the vector of rewards obtained by each skill, e.g., $skill = (r_{shoot}, r_{star})$ in the grid shooting environment. And the dissimilarity between two skill sets $skill_1, skill_2$ is defined as the value of their cross product $d_{skill}(skill_1, skill_2) = ||skill_1 \times skill_2||$. The dissimilarity metric is obtained by first sampling trajectories with different policies, and calculating the average dissimilarity between pair-wise skill sets, i.e., $Diversity_{skill} = \mathbb{E}_{skill_i, skill_j} d_{skill}(skill_i, skill_j)$ where $skill_i, skill_j$ are sampled from different styles.

2. **OT (optimal transport) metric** is based on the optimal transport distance (also known as Wasserstein distance). The similar idea is used in a recent work in imitation learning (Luo et al., 2023). When calculating the trajectory-level OT distance, we align steps with neighbor states together and sum up to the overall distance, i.e., $d_{OT}(\tau^1, \tau^2) = \arg\min_{\mu \in M} \sum_{t=1}^{T} \sum_{t'=1}^{T} ||s_t^1 - s_{t'}^2|| \mu_{t,t'}$ where $M = \{\mu \in \mathbb{R}^{T \times T}, \mu\mathbf{1} = \frac{1}{T}\mathbf{1}, \mu^T\mathbf{1} = \frac{1}{T}\mathbf{1}\}$. And the diversity metric is calculated by averaging all trajectory pairs $\tau^i, \tau^j$ from different policies $Diversity_{OT} = \mathbb{E}_{\tau^i, \tau^j} d_{OT}(\tau^i, \tau^j)$.
   We employ a normalization technique to enhance the interpretability of the OT metric. The normalization is achieved by applying the formula $normalize(Diversity_{OT}) = \frac{Diversity_{OT} - \alpha}{\alpha}$, where $\alpha$ is the OT metric on trajectories sampled from the same policy.

3. **Discrimination metric** represents how different policies can be discriminated by a neural network. After collecting trajectories of different policies, we first train a neural network to predict the policy index, and the discrimination metric is just the evaluation accuracy.

|  | Popularty metric | Skill metric | OT metric | Discrimination metric |
|---|---|---|---|---|
| SORL | 0.60 | 15.7 | 0.022 | 0.75 |
| SORL w/o advantage | 0.61 | 21.2 | 0.108 | 0.97 |
| Off-RLPMM | 0.68 | 23.2 | 0.048 | 0.99 |
| InfoGAIL | 0.00 | 2.0 | 0.000 | 0.69 |

Table 6: The diversity under differen metrics.

Based on the results of the additional diversity metrics presented in Table 6, we can conclude that the SORL algorithm is capable of obtaining diverse policies.

# G  COMPARISON WITH OFFLINE RL BASELINES

|  | SORL (ours) | CQL | AWR | IQL |
|---|---|---|---|---|
| SpaceInvaders | $387.1 \pm 33.3$ | $136.5 \pm 27.5$ | $339.3 \pm 57.1$ | $361.5 \pm 86.9$ |
| MsPacman | $622.2 \pm 65.3$ | $513.7 \pm 325.5$ | $486.7 \pm 148.1$ | $505.0 \pm 161.2$ |
| MontezumaRevenge | $306.7 \pm 19.1$ | $113.4 \pm 196.3$ | $343.3 \pm 173.9$ | $166.7 \pm 32.1$ |
| Enduro | $371.1 \pm 2.9$ | $0.0 \pm 0.1$ | $345.7 \pm 40.2$ | $290.1 \pm 44.2$ |
| Riverraid | $1931.2 \pm 432.1$ | $1127.7 \pm 160.6$ | $1710.3 \pm 270.0$ | $1909.3 \pm 332.1$ |
| Frostbite | $2056.0 \pm 391.9$ | $76.3 \pm 16.7$ | $1528.0 \pm 329.4$ | $2184.7 \pm 337.5$ |

Table 7: The comparision with offline RL baselines.

We present the results of standard offline RL methods in Table 7, which do not focus on diverse policy learning.

# H  ATARI ENVIRONMENT VISUALIZATION

## H.1  SPACEINVADERS

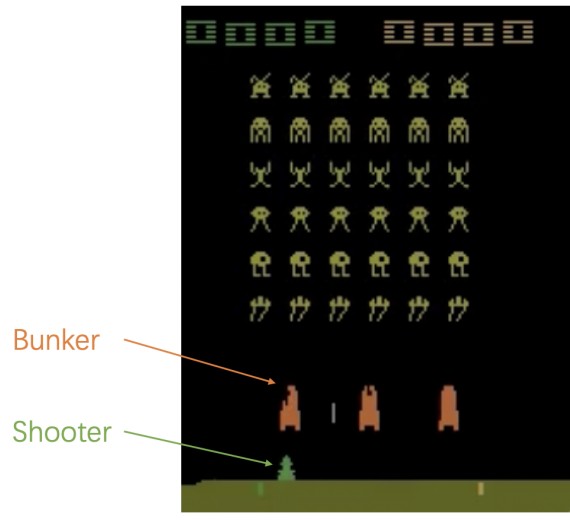

Figure 8: The game interface of SpaceInvaders.

In SpaceInvaders as Figure 8, the shooter can move horizontally, and there are three stationary bunkers positioned above the shooter. The shooter can shoot from bottom to top, in order to destroy the aliens above to get scores. During playing, there are random attacks from top bottom that can destroy the shooter. However, staying under the bunker can prevent those attacks. The bunker can also be destroyed by continuously attacking it. In addition, the aliens slowly swing left and right together all the time. In the game, players moves horizontally to shoot from a correct position with an alien above, and avoid being attacked.

|  | Leftmost - Bunker 1 | Bunker 1 - Bunker 2 | Bunker 2 - Bunker 3 | Bunker 3 - Rightmost |
|---|---|---|---|---|
| Style 1 | 23% | 19% | 36% | 22% |
| Style 2 | 10% | 57% | 15% | 18% |
| Style 3 | 32% | 38% | 26% | 4% |

Table 8: In SpaceInvaders, the probability of the shooter in different regions.

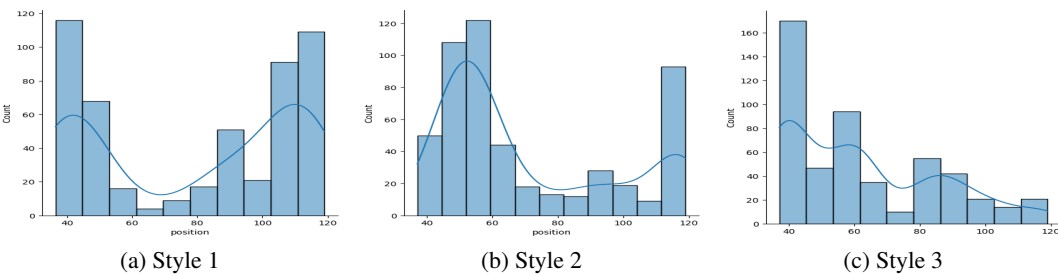

|  |  |  |
|:---:|:---:|:---:|
| (a) Style 1 | (b) Style 2 | (c) Style 3 |

Figure 9: In SpaceInvaders, the probability distribution of the shooter's position.

To evaluate the diversity of learned policies, we calculate the frequency of the shooter's position over 30,000+ steps for each style. Table 8 and Figure 9 demonstrates that different styles of policies tend to prefer different positions.

To better understand the style of learned clusters, we analyze the game mechanics which partially explains the preferences of styles. In SpaceInvaders, a shortcut of getting high scores is to staying close to the left of the Bunker 1 (and also staying close to the right of the Bunker 2) to destroy all the aliens in the leftmost column (and the rightmost column), because there are fewer attacks from top to bottom and is close to the bunker, where destroying the aliens in the leftmost/rightmost column provides a large amount to scores. As a result, style 1 and style 3 prefers staying in the leftmost, and style 1 and style 2 prefers staying in the leftmost.

## H.2 MsPacman

In MsPacman as shown in Figure 10, the map and starting position are fixed. To evaluate the diversity of learned policies, we calculate the frequency of the beginning trajectory until the first corner for each style, based on 30 trajectories per style. There are a total of six possible corners. Table 9 displays the different preferred corners for each style.

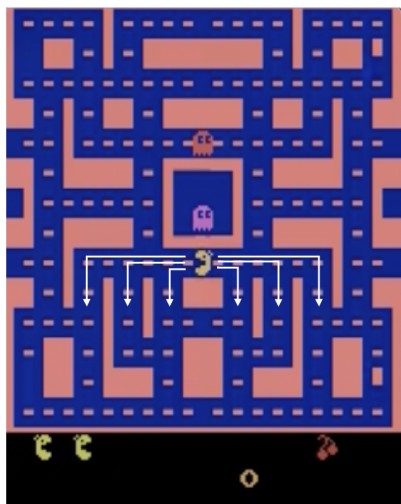

Figure 10: The game interface of MsPacman.

Besides, we analyze the visualizations of the dataset trajectories. Atari datasets in our experiment are collected by three players, which exhibit some diversity in terms of policy variations. As shown the following table 10, in the game MsPacman, different human players (named J, K and R) have varying preferences for the first visited corners. Additionally, even within a single player's gameplay, there can be variations in visitation patterns.

| Which corner | 3rd left | 2nd left | 1st left | 1st right | 2nd right | 3rd right |
|---|---|---|---|---|---|---|
| Style 1 | 0% | 33% | 67% | 0% | 0% | 0% |
| Style 2 | 20% | 7% | 7% | 3% | 37% | 27% |
| Style 3 | 0% | 97% | 3% | 0% | 0% | 0% |

Table 9: In MsPacman, the probability of the pacman choosing different routes in the beginning.

| Which corner | 3rd left | 2nd left | 1st left | 1st right | 2nd right | 3rd right |
|---|---|---|---|---|---|---|
| Player J (2 trajs) | 0% | 0% | 50% | 50% | 0% | 0% |
| Player K (2 trajs) | 0% | 0% | 50% | 50% | 0% | 0% |
| Player R (16 trajs) | 6% | 44% | 6% | 19% | 19% | 6% |

Table 10: In the dataset of MsPacman, the probability of the pacman choosing different routes in the beginning.

# I SENSITIVITY ANALYSIS ON THE NUMBER OF POLICIES

**Sensitivity analysis** We have performed experiments to evaluate the diversity, quality, and consistency of SORL under different values of $m$, where $m$ is the number of policies. The performance of SORL in Table 11 remains similar across different $m$ values, indicating that the algorithm is not highly sensitive to the hyperparameter $m$.

| | $m = 2$ | $m = 3$ | $m = 4$ | $m = 5$ |
|---|---|---|---|---|
| Diversity | 0.61 | 0.85 | 0.65 | 0.69 |
| Quality | 55.1% | 50.3% | 52.5% | 52.2% |
| Consistency | 86.4% | 88.5% | 88.7% | 88.7% |

Table 11: The diversity, quality and consistency under different values of $m$.

**The reward distribution of learned policies** The reward distribution of the learned policies is presented in Table 12. When taking larger $m$ value, the learned policies still differs a lot from each other. Some of them focus on shooting enemies, while others prefer collecting stars.

| $(m = 3)$ | Policy 1 | Policy 2 | Policy 3 | | |
|---|---|---|---|---|---|
| Reward (shoot) | 0.0 | 3.3 | 2.7 | | |
| Reward (star) | 8.2 | 0.4 | 3.8 | | |
| Winning rate | 59% | 36% | 56% | | |
| $(m = 4)$ | Policy 1 | Policy 2 | Policy 3 | Policy 4 | |
| Reward (shoot) | 1.3 | 2.2 | 0.2 | 3.1 | |
| Reward (star) | 6.0 | 3.2 | 7.6 | 0.5 | |
| Winning rate | 57% | 57% | 56% | 40% | |
| $(m = 5)$ | Policy 1 | Policy 2 | Policy 3 | Policy 4 | Policy 5 |
| Reward (shoot) | 1.1 | 2.6 | 2.9 | 0.4 | 1.5 |
| Reward (star) | 8.0 | 0.5 | 0.5 | 8.6 | 6.2 |
| Winning rate | 50% | 38% | 37% | 65% | 61% |

Table 12: The reward distribution when $m = 3, 4, 5$.

# J DISCUSSION ON POSTERIOR APPROXIMATION

In the main text, we approximate the true posterior $\hat{p}(z = i|\tau) \propto \int_{(s,a)\in\tau} \mu^{(i)}(a|s)$ with $\hat{p}(z = i|\tau) \propto \sum_{(s,a)\in\tau} \mu^{(i)}(a|s)$, because the consecutive multiplication leads to numerical instability.

We plot the distribution of $\hat{p}(z = 1|\tau)$ for all trajectories in the dataset in Figure 11. Figure 11a uses consecutive multiplication, while Figure 11b is the average of all steps. The results show that using

consecutive multiplication causes a trend of polarization in clustering, which causes unexpected numerical instability. The experiment was conducted in the grid shooting environment with the number of clusters set to $m = 2$.

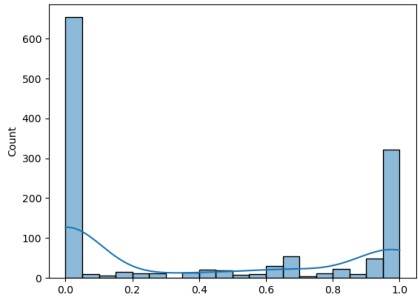 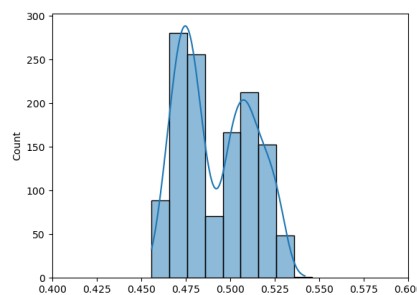

(a) The distribution of the true posterior $\hat{p}(z = 1|\tau)$, which is calculated by consecutive multiplication of the probability at each step.

(b) The distribution of the approximated $\hat{p}(z = 1|\tau)$, which is calculated by averaging the probability at each step.

Figure 11: The histogram of the distribution of the posterior.

