# OpenReview forum: "Stylized Offline Reinforcement Learning: Extracting Diverse High-Quality Behaviors from Heterogeneous Datasets"
_ICLR.cc/2024/Conference — ICLR 2024 poster_

### Official Review · Reviewer_BZGK · 2023-10-18

**Soundness:** 2 fair
**Presentation:** 3 good
**Contribution:** 3 good
**Rating:** 6
**Confidence:** 5

**Summary:**

This paper introduces an EM-inspired algorithm called Stylized Offline RL (SORL) to extract diverse strategies from heterogeneous offline RL datasets. Based on the learned behavior policies, the paper then applies an advantage-weighted style learning algorithm to improve their performance further. The authors demonstrated their algorithm's effectiveness with experiments on six Atari games and one online mobile basketball game, where SORL outperforms other baselines regarding quality, diversity, and consistency.

**Strengths:**

Extracting diverse behaviors from an offline RL dataset is an interesting problem. SORL efficiently solves the problem following an EM-based approach. The proposed evaluation criteria, considering quality, diversity, and consistency, provide a nice guideline for other researchers to follow. The algorithm also performs better than other existing baselines in multiple offline RL datasets, including the "Dunk City Diversity" dataset, which contains extremely diverse behaviors. Finally, the paper is overall well-written and easy to understand.

**Weaknesses:**

1. There is no theoretical ground for naively replacing $A^{\mu^{(i)}}$ with $A^\mu$ without importance sampling corrections. At least an empirical ablation study should be provided if it is difficult to devise a theoretical justification.

2. It isn't easy to understand the proof presented in Appendix B.

    (1) $\pi^{(i)}$ needs to satisfy the constraint $\int \pi^{(i)}(a\mid s)\,da=1$ for all $s$. The optimal solution might not be a critical point.

    (2) $A^\mu(s, a)-\lambda \mu^{(i)}(a\mid s)+\lambda \pi^{(i)}(a\mid s)+\lambda=0$ does not imply $\pi^{(i)*}(a\mid s)\propto \mu^{(i)}(a\mid s)\exp(\frac{1}{\lambda}A^\mu(s, a))$.

    (3) The normalization constant for $\pi^{(i)*}$ is ignored in (14).

3. The diversity metric proposed by the authors does not consider how different the styles are. For example, consider the case where $\pi^{(i)}(a=k\mid s)=\frac{1}{K}+\epsilon_k(s)$ where $K$ is the number of possible actions and $\epsilon_k(s)$ is a small number chosen arbitrarily. Then $\hat{p}(z=j\mid traj)$ would be determined by the values of $\epsilon_k(s)$, so $p_{popularity}$ would be close to a uniform distribution, which is the distribution that maximizes the entropy. However, the learned styles are far from being diverse.

### Minor comments:

1. How about using $\tau$ instead of $traj$? I think this notation is widely accepted.

2. The $-$ sign seems missing in (9).

3. §5.1 Off-RLMPP → Off-RLPMM (appears twice on the fourth line)

4. Appendix B: Unmatched parentheses $($ in (13) and on the first line of p.15

5. Appendix B: $exp$ → $\exp$ on the second line of p.15 and in the second and third equation of (14)

6. Appendix B: In (14), $t$ is the index of summation, but it does not appear in the summand.

7. Appendix B: Second $=$ → $\approx$

**Questions:**

1. §5.3 states that the character has to combat against an AI opponent. What is the strategy of the opponent? Is SORL robust to the changes in the opponent's strategy?

2. All experiments were conducted on environments with discrete action spaces. How does SORL perform in continuous action environments?

3. How was the diversity metric measured for InfoGAIL? To my knowledge, InfoGAIL does not explicitly split the policy into multiple clusters but instead learns a multi-modal policy.

4. In Appendix C, the paper states that

   > Besides, in order to ensure balanced learning among all the styles, we share the main network and use a lora module to discriminate different styles.

   The explanation on the "lora" module seems missing. Also, I recommend moving this part to the main paper.

---

> ### Author Response · Authors · 2023-11-20
> **(Part 1)**
>
> We greatly appreciate your thoughtful review. We have carefully considered the comments and have provided detailed clarification. We appreciate it if you have any further feedback.
>
> Q1: There is no theoretical ground for naively replacing $A_{\mu^{(i)}}$ with $A_\mu$ without importance sampling corrections. At least an empirical ablation study should be provided if it is difficult to devise a theoretical justification.
>
> A1: We conducted a comparison of performance (referred to as quality) using $A_{\mu^{(i)}}$ and $A_\mu$. We employ a soft clustering approach where each trajectory has a probability of belonging to each cluster. During the training of $A_{\mu^{(i)}}$, the dataset trajectories are weighted based on the probability obtained from the soft clustering method. In the experiment:
>
> |  | SpaceInvaders | MsPacman | MontezumaRevenge |
> | ---- | ---- | ---- | ---- |
> | SORL with $A_\mu$          | $387.1$ | $622.2$ | $306.7$ |
> | SORL with $A_{\mu^{(i)}}$ | $292.2$ | $551.3$ | $0.0$ |
>
> The empirical results indicate that utilizing $A_{\mu}$ during the learning process enhances the quality of policies. The reason that $A_\mu$ generally outperforms $A_{\mu^{(i)}}$ is that it only focus on part of the dataset.
>
> Q2: It isn't easy to understand the proof presented in Appendix B.
>
> A2: We sincerely appreciate the reviewer for pointing it out. We've refined the proof, which is now in Appendix D.
>
> Q3: The diversity metric proposed by the authors does not consider how different the styles are.
>
> A3: We appreciate the reviewer to point this out. We agree that one limitation of the current metric of diversity is that it relies on the clustering $\hat{p}(z=j|\tau)$. Comparing the diversity of two sets of policies are not trivial. Previous research, such as InfoGAIL, also did not propose a specific metric for evaluating diversity. Instead, they relied on visualization to showcase the diversity of learned policies.
> In order to provide a more comprehensive representation of the diversity of learned policies, we have proposed additional diversity metrics, under which the described problem does not exist. We have incorporated three additional metrics into our testing: the **skill metric**, the **OT (optimal transport) metric**, and the **discrimination metric**. The skill metric measures the dissimilarity of actions, while the OT metric quantifies the disparity in state distribution. The discrimination metric assesses whether the policies can be distinguished by a neural network. The table below showcases the diversity of different style learning methods in the grid shooting environment, measured using various metrics. Higher values indicate greater diversity. For a detailed description of these metrics, please refer to the Appendix F. Based on the results of the additional diversity metrics presented in Table below, we can conclude that the SORL algorithm is capable of obtaining diverse and high-quality policies. InfoGAIL fails to learn diverse policies, while Off-RLPMM has considerably lower winning rate.
>
> | | Popularity | Skill | OT | Discrimination | Winning rate |
> | ---- | ---- | ---- | ---- | ---- | ---- |
> | SORL                       | $0.60$ | $15.7$ | $29.14$ | $0.75$ | $55.1$% |
> | SORL w/o advantage | $0.61$ | $21.2$ | $31.61$ | $0.97$ | $45.2$% |
> | Off-RLPMM              | $0.68$ | $23.2$ | $29.87$ | $0.99$ | $40.4$% |
> | InfoGAIL                    | $0.00$ | $2.0$ | $28.53$ | $0.69$ | $55.3$% |
>
> Q4: §5.3 states that the character has to combat against an AI opponent. What is the strategy of the opponent? Is SORL robust to the changes in the opponent's strategy?
>
> A4: The opponent is an agent with a fixed strategy that move randomly, and shoot with a probability once the shooting action cools down.
>
> To check whether the agent is robust to the changes in the opponent's strategy, we introduce a moderately strong opponent with DQN training for 20,000 steps.
>
> |  | Reward (shoot) | Reward (star) | Winning rate |
> | ---- | ---- | ---- | ---- |
> | Policy 1 | $1.1$ | $0.11$ | $19$% |[数学公式]
> | Policy 2 | $0.1$ | $1.27$ | $54$% |
>
> Despite the opponent's increased strength, the policies employed by our agent remained stylized. Consequently, the winning rate experienced a decrease compared to previous opponents, but it remained competitive. Specifically, the winning rate of policy 1, which excels at shooting enemies, declined to $19$% due to the increased difficulty of successfully shooting and winning against the stronger opponent. On the other hand, the winning rate of policy 2, which specializes in collecting stars and focuses on escaping from the enemy's shooting, was not significantly affected.

---

> ### Author Response · Authors · 2023-11-20
> **(Part 2)**
>
> Q5: All experiments were conducted on environments with discrete action spaces. How does SORL perform in continuous action environments?
>
> A5: Although we don't include environment with continuous action space, our algorithm can be easily adapted to the continuous action space. The adaptation process involves representing action probability generated by the policy network as a continuous distribution (often Gaussian). Our method is not limited to discrete action space.
> The main contribution of SORL is to propose a novel framework that learns diverse and high-quaity qualities from offline datasets. The distinction between discrete and continuous action spaces in experiments lies outside the primary focus of this paper. We do recognize its importance for future work.
>
> Q6: How was the diversity metric measured for InfoGAIL? To my knowledge, InfoGAIL does not explicitly split the policy into multiple clusters but instead learns a multi-modal policy.
>
> A6: We set the latent space of InfoGAIL as an $m$-dim one-hot vector, where $m$ is the number of styles. For example, when $m=3$, we get the 3 policies using the one-hot latent codes $(0,0,1),(0,1,0),(1,0,0)$.
>
> Q7: The explanation on the "lora" module seems missing.
>
> A7: We appreciate the reviewer's advice to include a explanation of LoRA module [1] used. We have provided a more detailed explanation in Appendix E. We share the main network among policies of all styles and each policy has its own LoRA module.
>
> References:
>
> 1.Edward J. Hu, Yelong Shen, Phillip Wallis, Zeyuan Allen-Zhu, Yuanzhi Li, Shean Wang, Lu Wang, and Weizhu Chen. Lora: Low-rank adaptation of large language models, 2021

---

> > ### Comment · Reviewer_BZGK · 2023-11-22
> >
> > Thank you for your response. I still think the proof in Appendix D is incomplete. For example, the optimality of $\pi^{(i)}$ with respect to the Lagrangian does not automatically make it the optimal solution to the primal problem. Other than that, most of my concerns were resolved, so I raised my score.

---

> > > ### Author Response · Authors · 2023-11-23
> > >
> > > Thanks for increasing the score and further feedback.
> > >
> > > Q: The optimality of $\pi^{(i)}$ with respect to the Lagrangian does not automatically make it the optimal solution to the primal problem.
> > >
> > > A: The requirement of the extremum for the Lagrangian to be the optimal solution of the primal problem is enforcing the Karush-Kuhn-Tucker (KKT) condition. In the context of this optimization problem, the KKT condition can be derived from the Linear Independence Constraint Qualification (LICQ) condition. We will provide further details afterwards.

---

### Official Review · Reviewer_HX1W · 2023-10-30

**Soundness:** 4 excellent
**Presentation:** 3 good
**Contribution:** 3 good
**Rating:** 6
**Confidence:** 4

**Summary:**

This paper discusses the problem of extracting diverse as well as high-quality policies from multi-modal datasets via offline reinforcement learning. The core of the proposed method lies in clustering trajectories within the dataset. Behavior policyies to induce such clusters are learned, which are later used for constraining policy learning to ensure that the offline RL policies are high-performing as well as aligning with the diverse multi-modal dataset. Extensive experiments are conducted, and results seem positive. But I still have some concerns for this paper, please refer to the weaknesses.

**Strengths:**

1. The proposed method is straightforward and easy to comprehend.
2. Extensive experiments are conducted. Resutls on all three benchmarks show the proposed method SORL achieves balance between performance and diversity of the learned policies.
3. Procedure of SORL is clearly described.

**Weaknesses:**

1. Transformation from the true posterior to Eq. 2 needs more explanation. The current context is too weak. And I assume the basic assumption for this transformation is that all behavior policies $\mu_{1,..,m}$ are diverse enough, because the authors use transtion-wise action probability to replace the trajectory probability. This makes sense if behavior policies are diverse enough that they take different actions for each step. But if the policies only slightly differ from each other, the consecutive multiplication of all steps within the trajectory will make their trajectory distributions very different from each other (while the action distribution is not much different). As a result, Eq. 2 provides very inaccurate estimation of the posterior.
2. The proposed SORL needs to know the number $m$ of policy primitives constituting the dataset in order to learn the diverse policies. But it is hard to know this prior under many ciucumstances. I think there should be a study about how sensitive SORL is  to this hyperparameter.
3. How diverse are the induced policies? The case studies are great but there should be a quantitive study. My further question on this is, if we set $m$ larger than the actual number of policy primitives $\mu_{1,..,m}$, what will the resulted policies be like?
4. Some typos, e.g. line 10 in Algorithem 1: $\mu^{i}$ instead of $\mu^{1}$
5. As the author claims the induced policies are high-performing, the baselines should include some strong offline RL methods for comparison. This will also show SORL's advantage in policy diversity compared to them. The current baselines are too few.
6. How do the authors collect online user data? Where is the user agreement to collect this data? This should appear at least in the appendix.

**Questions:**

Please refer to the weaknesses. If the authors can address my concerns, I'm happy to increase my score.

---

> ### Author Response · Authors · 2023-11-20
>
> We greatly appreciate the valuable feedback provided by the reviewer. We have carefully considered the comments and have addressed them in our responses. If you have any additional questions or concerns, please don't hesitate to let us know.
>
> Q1: Transformation from the true posterior to Eq. 2 needs more explanation.
>
> A1: We appreciate the reviewer for pointing out that 'if the policies only slightly differ from each other, the consecutive multiplication of all steps within the trajectory will make their trajectory distributions very different from each other'. We plot the distribution of $\hat{p}(z=1|\tau)$ for all trajectories in the dataset in Figure 11, Appendix J. Figure 11(a) uses consecutive multiplication, while Figure 11(b) is the average of all steps. The results show that using consecutive multiplication causes a trend of polarization in clustering, which leads to unexpected numerical instability. The experiment was conducted in the grid shooting environment with the number of clusters set to $m=2$. To address this issue, we have chosen to average all the steps instead of multiplication in order to alleviate the problem of numerical instability.
>
> Q2: The proposed SORL needs to know the number of policy primitives constituting the dataset in order to learn the diverse policies. But it is hard to know this prior under many ciucumstances. I think there should be a study about how sensitive SORL is to this hyperparameter.
>
> A2: We appreciate the reviewer for pointing out the importance of conducting sensitivity analysis on the number of policies $m$ in SORL, as it is difficult to know the exact value of $m$ in real circumstances. In order to address this concern, we have conducted experiments to evaluate the diversity, quality, and consistency of SORL under different values of $m$. We also add two more diversity metrics that are unrelated to $m$, and their detailed descriptions can be found in Appendix F. The performance of SORL remains similar across different $m$ values, indicating that the algorithm is robust to the hyperparameter $m$.
>
> |  | $m=2$ | $m=3$ | $m=4$ | $m=5$ |
> | ---- | ---- | ---- | ---- | ---- |
> | Diversity       | $0.61$      | $0.85$     | $0.65$      | $0.69$ |
> | Diversity (skill metric) | $15.7$ | $20.3$ | $14.0$ | $14.5$ |
> | Diversity (OT metric)   | $29.1$ | $28.3$ | $28.5$ | $29.2$ |
> | Quality          | $55.1$% | $50.3$% | $52.5$% | $52.2$% |
> | Consistency | $86.4$% | $88.5$% | $88.7$% | $88.7$% |
>
> Q3:  How diverse are the induced policies? The case studies are great but there should be a quantitive study. My further question on this is, if we set  m  larger than the actual number of policy primitives  $\mu_{1,\cdots,m}$ , what will the resulted policies be like?
>
> A3: The quantitative results of the grid shooting environment can be found in the table provided below (also available in Appendix A).
>
> |  | Quality | Diversity | Consistency |
> | ---- | ---- | ---- | ---- |
> | SORL          | $55.1\pm4.6$% | $0.60\pm0.07$ | $86.4\pm0.4$% |
> | Off-RLPMM | $40.4\pm0.7$% | $0.68\pm0.01$ | $90.5\pm0.0$% |
> | InfoGAIL      | $55.3\pm2.1$% | $0.00\pm0.00$ | $84.0\pm0.3$% |
>
> We provide additional results in the grid shooting game as $m$ increases. The actual number of policy primitives is two. The reward distribution of the learned policies is presented in the following table, taking $m=4$ as an example. For brevity, the resulting policies for $m=3$ and $m=5$ can be found in Appendix I.
>
> | Learned policies | Policy 1 | Policy 2 | Policy 3| Policy 4 |
> | ---- | ---- | ---- | ---- | ---- |
> | Reward (shoot) | $1.3$     | $2.2$     | $0.2$    | $3.1$ |
> | Reward (star)   | $6.0$    | $3.2$     | $7.6$    | $0.5$ |
> | Winning rate    | $57$% | $57$% | $56$% | $40$% |
>
> When taking larger $m$ values, the learned policies still differs a lot from each other. In the case when $m=4$, one policy learns to focus on shooting, one policy concentrates on star collection, and the remaining policies' preferences lie between the above two extremes.

---

> ### Author Response · Authors · 2023-11-20
> **(Part 2)**
>
> Q4: As the author claims the induced policies are high-performing, the baselines should include some strong offline RL methods for comparison. This will also show SORL's advantage in policy diversity compared to them. The current baselines are too few.
>
> A4: We add experimental results of standard offline RL methods (CQL, AWR and IQL). We conduct comparison in Atari games and results are shown below. Notably, SORL learns a set of policies while the remaining methods learn a single policy. The diversity metric cannot be calculated for methods that only generate a single policy. The scores of SORL are averaged over the full set of learnt policies. We have not yet completed all the experiments on AWR. However, we will finish them and include them in Appendix G before the deadline.
>
> |        | SORL | CQL | AWR | IQL |
> | ---- | ----     | ----   | ----   | ----  |
> | SpaceInvaders            | $387.1\pm33.3$    | $136.5\pm27.5$   | $351.3\pm102.6$ | $361.5\pm86.9$ |
> | MsPacman                  | $622.2\pm65.3$    | $513.7\pm325.5$  | $320.0\pm68.53$ | $505.0\pm161.2$ |
> | MontezumaRenvenge | $306.7\pm19.1$    | $113.4\pm196.3$  | $210.0\pm119.9$ | $166.7\pm32.1$ |
> | Enduro                        | $371.1\pm2.9$       | $0.0\pm0.1$          | $301.7\pm66.1$ | $290.1\pm44.2$ |
> | Riverraid                     | $1931.2\pm432.1$ | $1127.7\pm160.6$ | $1439.0\pm280.6$ | $1909.3\pm332.1$ |
> | Frostbite                     | $2056.0\pm391.9$ | $76.3\pm16.7$     | $1165.0\pm470.6$ | $2184.7\pm377.5$ |
>
> The results above shows that the quality performance of SORL is still generally higher than offline RL baselines. We hypothesize that the human datasets are multi-modal, and utilizing multiple policies instead of a single policy to fit the dataset yields better results.
>
> Q5: How do the authors collect online user data? Where is the user agreement to collect this data? This should appear at least in the appendix.
>
> A5: The game environment and the dataset used in this paper are public. The link to the game environment is <https://github.com/FuxiRL/DunkCityDynasty>, and the link to the dataset is <https://huggingface.co/datasets/FUXI/DunkCityDynasty_Dataset>. To our best knowledge, this paper is the first to conduct experiments of learning both diverse and high-quality behaviors from a large human dataset of video games.

---

> ### Comment · Reviewer_HX1W · 2023-11-22
>
> Many thanks for conducting the additional experiments. My concerns have been addressed and I have increased my score.

---

> > ### Author Response · Authors · 2023-11-23
> >
> > Thanks for raising the score. We are pleased that the quality of this work has been enhanced due to your insightful questions and suggestions.

---

### Official Review · Reviewer_Xp9f · 2023-10-31

**Soundness:** 3 good
**Presentation:** 2 fair
**Contribution:** 3 good
**Rating:** 6
**Confidence:** 5

**Summary:**

This paper presents a new approach, Stylized Offline RL (SORL), which seeks to derive high-quality, stylistically diverse policies from offline datasets with distinct behavioral patterns. While most reinforcement learning (RL) methodologies prioritize either online interactions or policy performance, SORL combines the Expectation-Maximization (EM) algorithm with trajectory clustering and advantage-weighted style learning to promote policy diversification and performance enhancement. Through experiments, SORL has been shown to outperform previous methods in generating high-quality and diverse policies, with a notable application being in the basketball video game "Dunk City Dynasty". The effectiveness of SORL is evaluated in various settings, including a basketball video game. Compared to other methods, SORL consistently yields better-performing policies that also maintain distinct behavior patterns. The paper's contributions include:
* The introduction of SORL, a framework that combines quality and diversity into the optimization objective, addressing limitations in both diverse RL and offline RL methods.
* Extensive evaluations showing SORL's ability to generate high-quality, stylistically diverse policies from diverse offline datasets, including human-recorded data.

**Strengths:**

1.	The structure of the paper is clear and easy to understand.
2.	The idea of using the EM framework to do trajectory clustering and policy optimization is interesting.
3.	Using offline RL to solve real-world tasks using a human-generated dataset shows the scalability of the proposed method.

**Weaknesses:**

1.	No standard Offline RL baseline compared. As quality and diversity are both metrics for evaluation, it would be good to compare the performance with other standard offline RL methods, e.g., CQL, TD3+BC, AWR.
2.	The motivation for increasing the diversity of policy is not clear. In the related work section, the authors only discuss the importance of diversity in online RL settings, for example, encouraging exploration, better opponent modeling, and skill discovery. However, in the offline RL setting, there is no exploration problem or skill discovery since the dataset is fixed. In addition, in the preliminary section, the authors aim to “learn a set of high-quality and diverse policies” without any explanation of the advantage of learning a set of diverse policies over a single policy with diverse behaviors (e.g., using multi-modal distribution as policy distribution).
3.	Many details are missing in the experiment of the “Dunk City Dynasty”. The code does not include this experiment.

**Questions:**

1.	The metric for evaluating diversity seems to rely on the learned clustering p. I wonder why don’t evaluate the diversity of the learned policy? Otherwise, this metric cannot be used for algorithms that don’t learn the clustering of datasets. In addition, the goal of clustering the dataset is to learn diverse policies for online evaluation, so the diversity of the policy is what we really care about.
2.	Could the authors provide some visualization or example of the mean of the clusters in Atari games? It is not intuitive how the diverse behavior looks like in those games. Similarly, in the Dunk City Dynasty game, the visualization in Figure 3 is too simple. Could the authors plot the shooting positions of each policy? Also, besides the shooting position, are there other differences between these policies?
3.	Could the authors provide more details about the setting of the Dunk City Dynasty experiments? For example, the action space, and model structure. Appendix C only describes the structure of the first two experiments.
4.	What does it mean by “we share the main network and use a lora module to discriminate different styles.” In Appendix C?
5.	Results in Table 3 show that SORL has a very high variance (5.3 ± 3.4) in terms of quality, which only slightly outperforms InfoGAIL (5.0 ± 0.8). Does this mean pursuing high-quality sacrifices for the performance of the policy? Then, what do we gain from the high diversity?

---

> ### Author Response · Authors · 2023-11-20
> **(Part 1)**
>
> Thanks for the valuable feedback. We appreciate the reviewer's constructive comments and would like to address your concerns. Please let us know if you have any further questions or comments.
>
> Q1: No standard Offline RL baseline compared. As quality and diversity are both metrics for evaluation, it would be good to compare the performance with other standard offline RL methods, e.g., CQL, TD3+BC, AWR.
>
> A1: We add experimental results of standard offline RL methods (CQL, AWR and IQL). We exclude TD3+BC, because it works for the continuous action setting, different from the experiment settings with discrete action spaces. Instead, we include IQL (Offline Reinforcement Learning with Implicit Q-Learning), which is a strong offline RL algorithm. We conduct comparison in Atari games and results are shown below. Notably, SORL learns a set of policies while the remaining methods learn a single policy. The scores of SORL are averaged over the full set of learnt policies. We have not yet completed all the experiments on AWR. However, we will finish them and include them in Appendix G before the deadline.
>
> |        | SORL | CQL | AWR | IQL |
> | ---- | ----     | ----   | ----   | ----  |
> | SpaceInvaders            | $387.1\pm33.3$    | $136.5\pm27.5$   | $351.3\pm102.6$ | $361.5\pm86.9$ |
> | MsPacman                  | $622.2\pm65.3$    | $513.7\pm325.5$  | $320.0\pm68.53$ | $505.0\pm161.2$ |
> | MontezumaRenvenge | $306.7\pm19.1$    | $113.4\pm196.3$  | $210.0\pm119.9$ | $166.7\pm32.1$ |
> | Enduro                        | $371.1\pm2.9$       | $0.0\pm0.1$          | $301.7\pm66.1$ | $290.1\pm44.2$ |
> | Riverraid                     | $1931.2\pm432.1$ | $1127.7\pm160.6$ | $1439.0\pm280.6$ | $1909.3\pm332.1$ |
> | Frostbite                     | $2056.0\pm391.9$ | $76.3\pm16.7$     | $1165.0\pm470.6$ | $2184.7\pm377.5$ |
>
> The results above show that the quality performance of SORL is still generally higher than offline RL baselines. We hypothesize that this is because the human datasets are multi-modal, and utilizing multiple policies instead of a single policy to fit the dataset yields better results.
>
> Q2: The motivation for increasing the diversity of policy is not clear.
>
> A2: We have discussed the motivation of learning diverse policies from offline datasets in Lines 4-7 in the introduction. Additionally, we have added further content discussing the utilization of learning diverse policies from offline datasets in opponent modeling in Lines 8-10. We appreciate the reviewer for pointing out that the motivation needs a more detailed discussion. Learning diverse policies from offline datasets is beneficial for various real-world applications, such as game AI and opponent modeling. In game AI, offline data collected from human gameplay allows AI bots to mimic human-like behavior. In online games, deploying such AI bots with varyied motion styles can enrich the gaming environment and enhance player engagement. Additionally, in opponent modeling, high-quality diverse opponents that resemble real opponents can significantly improve the performance of the learned policy. It is worth noting that learning diverse policies from offline datasets is an area that has not been extensively explored in previous research. In the related work, we mention encouraging exploration and skill discovery because they are the motivation of promoting diversity in online settings rather than offline settings.
>
> Q3: Many details are missing in the experiment of the “Dunk City Dynasty”. The code does not include this experiment.
>
> A3: We have updated the supplementary materials for "Dunk City Dynasty" and included the algorithm codes. However, please note that the code for large-scale efficient experiments cannot be made public at this time due to institution regulations. We will make it available to the public as much as possible in the future.

---

> ### Author Response · Authors · 2023-11-20
> **(Part 2)**
>
> Q4: The metric for evaluating diversity seems to rely on the learned clustering p.
>
> A4: We admit that the diversity metric employed in the paper relies on clustering $p$. Comparing the diversity of two sets of policies are not trivial. Previous research, such as InfoGAIL, also did not propose a quantitative metric for evaluating diversity when learning policies from datasets. Instead, they relied on visualization to showcase the diversity of learned policies.
>
> In order to provide a more comprehensive investigation of the diversity of learned policies, we have proposed additional diversity metrics that do not require clustering: the **skill metric** measuring the dissimilarity of the skill set and the **OT (optimal transport) metric** quantifing the disparity in state distribution. For a detailed description of these metrics, please refer to the Appendix F. Moreover, we will present visualization results of the Atari environment in the response to the next question.
>
> The table provided below showcases the diversity of various style learning methods in the grid shooting environment utilizing these two additional metrics as well as popularity metric used in the original paper. The two new metrics are not dependent on clustering. Higher values indicate greater diversity. Based on the results of the additional diversity metrics presented in Table below, we can conclude that the SORL algorithm is capable of obtaining diverse policies. The results demonstrate that SORL learns a set of diverse and high-quality policies. InfoGAIL fails to learn diverse policies, while Off-RLPMM learns low-quality diverse policies and has considerably lower winning rate.
>
> | | Popularity metric | Skill metric | OT metric | Winning rate |
> | ---- | ---- | ---- | ---- | ---- |
> | SORL                      | $0.60$ | $15.7$ | $29.14$ | $55.1$% |
> | SORL w/o advantage | $0.61$ | $21.2$ | $31.61$ | $45.2$% |
> | Off-RLPMM              | $0.68$ | $23.2$ | $29.87$ | $40.4$% |
> | InfoGAIL                     | $0.00$ | $2.0$ | $28.53$ | $55.3$% |
>
> Q5: Could the authors provide some visualization or example of the mean of the clusters in Atari games? Could the authors plot the shooting positions of each policy? Also, besides the shooting position, are there other differences between these policies?
>
> A5: We have generated visualizations of the clusters in several Atari environments. Detailed descriptions of the environment and the visualization can be found in Appendix H. In SpaceInvaders, the shooter can move horizontally, and there are three stationary bunkers positioned above the shooter. To evaluate the diversity of learned policies, we calculate the frequency of the shooter's position over 30,000+ steps for each style. The table below demonstrates that different styles of policies tend to prefer different positions.
>
> |  | Leftmost - Bunker 1 | Bunker 1 - Bunker 2 | Bunker 2 - Bunker 3 | Bunker 3 - Rightmost|
> | ---- | ---- | ---- | ---- | ---- |
> | Style 1 | $23$% | $19$% | $36$% | $22$% |
> | Style 2 | $10$% | $58$% | $15$% | $18$% |
> | Style 3 | $32$% | $38$% | $26$% | $4$% |
>
> In MsPacman, the map and starting position are fixed. To evaluate the diversity of learned policies, we calculate the frequency of the beginning trajectory until the first corner for each style, based on 30 trajectories per style. There are a total of six possible corners. The table below displays the different preferred corners for each style:
>
> | Which corner | 3rd left | 2nd left | 1st left | 1st right | 2nd right | 3rd right |
> | ---- | ---- | ---- | ---- | ---- | ---- | ---- |
> | Style 1 | $0$% | $33$% | $67$% | $0$% | $0$% | $0$% |
> | Style 2 | $20$% | $7$% | $7$%   | $3$% | $37$% | $27$% |
> | Style 3  | $0$% | $97$% | $3$%   | $0$%  | $0$%  | $0$% |
>
> As for the Dunk City Dynasty, the plot of shooting positions are added to Appendix B. Besides the style difference of shooting positions, there also exists the style difference of ball passing and ball possesion, as shown in the following table. Style 2 has the fewest number of passes and the lowest proportion of goals scored directly after a pass.
>
> |  | Style 1 | Style 2 | Style 3 |
> | ---- | ---- | ---- | ---- |
> | #passes | $41$ | $29$ | $39$ |
> | Proportion of goals directly after a pass | $27.7$% | $22.2$% | $44.4$% |

---

> ### Author Response · Authors · 2023-11-20
> **(Part 3)**
>
> Q6: Could the authors provide more details about the setting of the Dunk City Dynasty experiments? For example, the action space, and model structure. Appendix C only describes the structure of the first two experiments.
>
> A6: We appreciate the reviewer for pointing it out. We have provided details of the experiments in Dunk City Dynasty in Appendix B. The state space dimension is $468$, including global state, allies' states and enemies' states. The action space is a discrete space with $52$ actions. The network structure of 'Dunk City Dynasty' is a 3-layer MLP. The opponent in evaluation is an agent with moderate strength, that is learned by vanilla Behavioral Cloning for similar training steps (100,000 steps).
>
> Q7: What does it mean by “we share the main network and use a lora module to discriminate different styles.” In Appendix C?
>
> A7: The detailed description of LoRA module is as follows. In order to ensure balanced learning among all the styles, we share the main network and use a LoRA module to discriminate different styles. LoRA [1] is a widely used network structure, that substitutes the original matrix of the linear layer by a matrix and a low-rank multiplication of two other matrices. Further details of LoRA module is added in Appendix E.
>
> Q8: Results in Table 3 show that SORL has a very high variance (5.3 ± 3.4) in terms of quality, which only slightly outperforms InfoGAIL (5.0 ± 0.8). Does this mean pursuing high-quality sacrifices for the performance of the policy? Then, what do we gain from the high diversity?
>
> A8: The focus of our work is to learn diverse and high-quality policies. These are two distinct objectives and pursuing diversity may sacrifice the quality of the policy. The goal of this paper is to maximize both objectives and trade off between diversity and quality. In Table 3, although SORL shows higher variance in performance, it signficantly outperforms the baselines in terms of disversity. Similarly, in the other two environments, we also observe a trade-off between quality and diversity. For instance, Figure 2 in the main text shows that the SORL algorithm does not always outperform the baselines in terms of quality, but it consistently achieves a good balance between high quality and diversity. By combining the pursuit of both quality and diversity, we demonstrate that the SORL algorithm is superior to the baselines. It provides a more comprehensive solution that achieves both high performance and a diverse set of policies.
>
> The motivation of learning diverse policies from offline datasets has been expained in the response to question 2. To recapitulate briefly, it is useful in applications including learning human-like behaviors in game AI and opponent modeling.
>
> References
>
> 1.Edward J. Hu, Yelong Shen, Phillip Wallis, Zeyuan Allen-Zhu, Yuanzhi Li, Shean Wang, Lu Wang, and Weizhu Chen. Lora: Low-rank adaptation of large language models, 2021.

---

> > ### Comment · Reviewer_Xp9f · 2023-11-21
> > **Follow-up questions**
> >
> > I really appreciate the efforts made by the authors. The experimental results and explanations added in the rebuttal stage make this paper much better. **Actually, I feel that the version after the rebuttal looks more like a formal submission to ICLR while the previous version seems incomplete with only preliminary results.** However, I am not sure if this is fair to other submissions as the rebuttal phase should only be used for clarifying unclear questions rather than completing the paper.
> >
> > I still have some follow-up questions:
> >
> > Q1: “We hypothesize that this is because the human datasets are multi-modal, and utilizing multiple policies instead of a single policy to fit the dataset yields better results.” Could the authors provide more evidence for this hypothesis? For example, the performance of individual policies. Actually, I also want to ask why using the Atari game to evaluate the proposed method. Although it is a widely used benchmark for offline RL, I am unsure if it is reasonable to say it is a heterogeneous dataset. Why not use a real heterogeneous dataset and show that a diverse policy works well? I suggest that this paper should only focus on the Dunk City Dynasty experiment, which seems to be more consistent with the motivation and complex enough to compare different methods.
> >
> >
> > Q2: Increasing diversity in online settings is indeed a widely investigated topic, but I am still not fully convinced to maximize diversity in offline settings. The authors give me two examples:
> > * (1)	in game AI, diverse behavior enriches the gaming environment and enhances player engagement.
> > * (2)	in opponent modeling, high-quality diverse opponents that resemble real opponents can significantly improve the performance of the learned policy.
> >
> > These two motivations seem reasonable to me on a general level, but both are hard to evaluate. I suggest that the authors design some examples to support the motivation. Or maybe just focus on the Dunk City Dynasty experiment and provide more analysis and ablation studies.
> >
> >
> > Q4: The proposed two metrics, skill metric, and OT metric, seem to have inconsistent results. Why does InfoGAIL have a similar OT score but a much lower skill score than SORL? Which metric leads to the conclusion “InfoGAIL fails to learn diverse policies”?

---

> > > ### Author Response · Authors · 2023-11-22
> > > **(Part 1) Response to Follow-up Questions**
> > >
> > > We would like to express our sincere appreciation to the reviewer for their insightful advice and positive feedback during the rebuttal. We are pleased that the reviewer acknowledged our efforts to "make this paper much better". We kindly request the reviewer to  re-evaluate our paper.
> > >
> > > **Q1-1:** “We hypothesize that this is because the human datasets are multi-modal, and utilizing multiple policies instead of a single policy to fit the dataset yields better results.” Could the authors provide more evidence for this hypothesis? For example, the performance of individual policies.
> > >
> > > **A1-1:** The performances of three clustered policies learned by SORL are $[260,358,408]$ (SpaceInvaders) and $[674,553,591]$ (MsPacman), based on 100 evaluations. And the visualization results of policy differences can be found in the response to **Q5** in our previous response.
> > >
> > > **Q1-2:** Actually, I also want to ask why using the Atari game to evaluate the proposed method. Although it is a widely used benchmark for offline RL, I am unsure if it is reasonable to say it is a heterogeneous dataset. Why not use a real heterogeneous dataset and show that a diverse policy works well? I suggest that this paper should only focus on the Dunk City Dynasty experiment, which seems to be more consistent with the motivation and complex enough to compare different methods.
> > >
> > > **A1-2:**  Atari datasets in our experiment are collected by three players [1], which do exhibit some diversity in terms of policy variations. For instance, as shown the following table, in the game MsPacman, different human players (named J, K and R) have varying preferences for the first visited corners. Additionally, even within a single player's gameplay, there can be variations in visitation patterns.
> > >
> > > | Which corner | 3rd left | 2nd left | 1st left | 1st right | 2nd right | 3rd right |
> > > | ---- | ---- | ---- | ---- | ---- | ---- | ---- |
> > > | Player J (2 trajs)    | $0$% | $0$% | $50$% | $50$% | $0$% | $0$% |
> > > | Player K (2 trajs)   | $0$% | $0$% | $50$% | $50$% | $0$% | $0$% |
> > > | Player R (16 trajs) | $6$% | $44$% | $6$% | $19$% | $19$% | $6$% |
> > >
> > > We agree with the reviewer's suggestion that more empirical studies can be done on Dunk City Dynasty which uses a real-world heterogeneous dataset to better demonstrate the effectiveness of diverse policies. The reason that we use Atari is that we believe it is still necessary to provide experiment results on an popular open-source environment. We think it helps to better position our research in the broader research community.
> > >
> > > **Q2:** These two motivations seem reasonable to me on a general level, but both are hard to evaluate. I suggest that the authors design some examples to support the motivation. Or maybe just focus on the Dunk City Dynasty experiment and provide more analysis and ablation studies.
> > >
> > > **A2:** We appreciate the reviewer for agreeing with the motivations. We have demonstrated the ability of the SORL algorithm to generate diverse policies with high quality through quantitative experiments, qualitative experiments, and visualization results. In Dunk City Dynasty, as depicted in Figure 3, we provide evidence that SORL learns policies with distinct shooting position preferences that are clearly perceivable by human.
> > >
> > > We agree with the reviewer that, in order to further investigate the influence of these diverse policies on human players' experiences and to consolidate the motivation of stylized offline RL, it is important to deploy the algorithm in real systems and to conduct human subject experiments. It is worth noting that fulfilling these requirements may pose challenges in terms of additional regularization and time constraints, particularly within a short period. Nevertheless, we acknowledge the importance of further exploring these aspects as an intriguing avenue for future research.

---

> > > ### Author Response · Authors · 2023-11-22
> > > **(Part 2) Response to Follow-up Questions**
> > >
> > > **Q4:** The proposed two metrics, skill metric, and OT metric, seem to have inconsistent results. Why does InfoGAIL have a similar OT score but a much lower skill score than SORL? Which metric leads to the conclusion “InfoGAIL fails to learn diverse policies”?
> > >
> > > **A4:** To clarify this observation in the results between the skill metric and OT metric, we conducted an analysis. We calculated the OT metric on trajectories sampled from the same policy, denoted as  $\alpha=28.52$. In the case of InfoGAIL, the calculated OT metric is $28.53$, which indicates minimal diversity according to this metric. This observation aligns with the low skill metric. Therefore, both the OT metric and the skill metric support the conclusion that InfoGAIL fails to learn diverse policies.
> > >
> > > In order to mitigate confusion caused by the unscaled values of the OT metric, we performed a normalization process. The normalization is achieved by applying the formula $normalize(OT)=\frac{OT-\alpha}{\alpha}$, where $\alpha$ is the OT metric on trajectories sampled from the same policy. Based on this updated table, we can draw the conclusion that the SORL algorithm learns a set of diverse and high-quality policies. InfoGAIL fails to learn diverse policies, while Off-RLPMM learns low-quality diverse policies and has considerably lower winning rate.
> > >
> > > | | Popularity metric | Skill metric | OT metric (normalized) | Winning rate |
> > > | ---- | ---- | ---- | ---- | ---- |
> > > | SORL                      | $0.60$ | $15.7$ | $0.022$ | $55.1$% |
> > > | SORL w/o advantage | $0.61$ | $21.2$ | $0.108$ | $45.2$% |
> > > | Off-RLPMM              | $0.68$ | $23.2$ | $0.048$ | $40.4$% |
> > > | InfoGAIL                     | $0.00$ | $2.0$ | $0.000$ | $55.3$% |
> > >
> > > References:
> > > 1.Zhang, R., Walshe, C., Liu, Z., Guan, L., Muller, K., Whritner, J., Zhang, L., Hayhoe, M., & Ballard, D. (2020). Atari-HEAD: Atari Human Eye-Tracking and Demonstration Dataset. Proceedings of the AAAI Conference on Artificial Intelligence, 34(04), 6811-6820.

---

> > > > ### Comment · Reviewer_Xp9f · 2023-11-22
> > > > **Response to authors**
> > > >
> > > > Thanks to the authors for providing more evidence to support their points. I don't have further questions and I think the discussion so far has provided enough ingredients to make this paper a good one. I tend to accept this paper now and will increase my score. However, I highly suggest the authors spend some effort to organize the results and clarification during the rebuttal stage to improve the quality of the manuscript.

---

> > > > > ### Author Response · Authors · 2023-11-23
> > > > >
> > > > > Thank you for giving valuable questions and suggestions, which have enhanced the quality of this work. And thanks for raising the score. We will certainly incorporate the results during the rebuttal stage into the manuscript.

---

### Official Review · Reviewer_pT1t · 2023-11-02

**Soundness:** 3 good
**Presentation:** 3 good
**Contribution:** 3 good
**Rating:** 8
**Confidence:** 4

**Summary:**

The paper addresses the problem of learning diverse policies based on datasets of trajectories collected by humans. This is particularly relevant in the context of video gaming, where the goal is to develop bots that are not only proficient but also exhibit varied behavioral patterns based on human player data. The authors introduce a purely offline solution that eliminates the need for environmental interaction. This approach is underpinned by a dual-step method. Initially, a clustering technique, leveraging the EM algorithm, assigns trajectories to different clusters by learning  a style-sensitive policy. Subsequently, to foster policies that are both effective and stylistically aligned, Advantage Weighted Regression (AWR) is employed in conjunction with a style-regularization component based on the style-sebsitive policies. The effectiveness of this method is demonstrated through a series of tests conducted in a simplistic environment, a handful of Atari games, and a commercial video game, all of which confirm the algorithm's capability to generate diverse and competent policies.

**Strengths:**

The paper is well written and will be of the interest of a large audience. The model is quite simple (clustering then offline learning) and easy to apply to different use-cases, it can be a good baseline for many future works.  More importantly, as far as I know, this paper is the first one to propose a set of experiments on a real video game and a large dataset of collected traces which is certainly where this paper has the most value and the dataset and  environment will be release (can you confirm I am right on that point ?)

**Weaknesses:**

The paper presents a compelling methodology, yet it notably omits a benchmark against "robust imitation of diverse behaviors," which is a reference work within this domain. Although primarily an online training paper, like infoGAIL, its principles could potentially be adapted for offline training, serving as a relevant comparison.

There appears to be an implied relationship between what the authors denote as 'style' and the rewards associated with a particular trajectory. Commonly, one might categorize trajectories by skill level, segregating expert from intermediate or novice plays. However, in such a scenario, the operation of the Advantage Weighted Regression (AWR) on these distinct clusters is not thoroughly explained. The connection between the 'style' of play and the 'reward' outcome merits a deeper examination.

The simplicity of the clustering model raises concerns regarding its ability to discern more nuanced styles, such as specific repetitive actions (e.g., "jump twice"). A more critical discussion on the model's capacity to identify and differentiate between complex styles would enhance the paper.The algorithm seems limited in capturing policies that would need memory to characteriwe their styles.

Regarding the implementation of AWR, it seems to be applied to each cluster individually. This approach suggests that in a situation where ten clusters are identified, only one-tenth of the training trajectories are utilized during the AWR phase for each cluster. This potentially limits the method's scalability when dealing with numerous styles, possibly making it impractical for extensive style differentiation.

Lastly, the paper could explore the potential of employing more advanced offline reinforcement learning algorithms in the second step of the methodology. Such a discussion could provide insights into improving the efficiency and effectiveness of the learning process in diversifying policies.

**Questions:**

(see previous section)

**Details Of Ethics Concerns:**

no concerns

---

> ### Author Response · Authors · 2023-11-20
>
> We thank the reviewer for the constructive comments, and we provide clarification on your concerns as follows. We would appreciate it if you have any further feedback.
>
> Q1: Will the video game environment and the dataset be released?
>
> A1: The game environment and the dataset used in this paper are public. The link to the game environment is <https://github.com/FuxiRL/DunkCityDynasty>, and the link to the dataset is <https://huggingface.co/datasets/FUXI/DunkCityDynasty_Dataset>. To our best knowledge, this paper is the first to conduct experiments of learning both diverse and high-quality behaviors from a large human dataset of video games.
>
> Q2: The connection between the 'style' of play and the 'reward'.
>
> A2: We agree with the reviewer that 'the connection between the 'style' of play and the 'reward' outcome merits a deeper examination'. In our perspective, there are two types of policy diversity: different policies of comparable performances and policies of novice/veteran distinction. In the first case, although the policies are diverse, they may still have similar expected returns. In this paper, we primarily focus on this case. For the latter case, one simple approach is to split the dataset into subsets with respect to the cumulative return of trajectories, and learn different policies on corresponding subsets. In the experiments (Section 5.3), we find that our method discovers two distinct strategies for winning the grid shooting game: shooting the enemy and collecting stars. The motivation of this paper is to enhance the quality concerning the first kind of diversity.
>
> Q3: The simplicity of the clustering model raises concerns regarding its ability to discern more nuanced styles. A more critical discussion on the model's capacity to identify and differentiate between complex styles would enhance the paper.
>
> A3: Regarding to your concerns on its ability to discern more nuanced styles, we acknowledge that the MLP utilized in our current implementation cannot model policies that need memory to characterize styles, since it doesn't memorize historical states. We believe that future work could incoperate network structures with memory, such as transformers and recurrent neural networks (RNNs).
>
> To discuss more on the model's capacity to identify and differentiate between complex styles, we provide an additional experiment on different numbers of styles $m$, especially when $m$ is set larger than the actual number of styles of the dataset.
>
> We have conducted experiments to evaluate the diversity, quality, and consistency of SORL under different values of $m$. We also add two more diversity metrics that are unrelated to $m$, and their detailed descriptions can be found in Appendix F. The results show that the performance of SORL remains consistent across different values of $m$, indicating that the algorithm has sufficient capacity to generate diverse policies.
>
> |  | $m=2$ | $m=3$ | $m=4$ | $m=5$ |
> | ---- | ---- | ---- | ---- | ---- |
> | Diversity       | $0.61$      | $0.85$     | $0.65$      | $0.69$ |
> | Diversity (skill metric) | $15.7$ | $20.3$ | $14.0$ | $14.5$ |
> | Diversity (OT metric)   | $29.1$ | $28.3$ | $28.5$ | $29.2$ |
> | Quality          | $55.1\%$ | $50.3\%$ | $52.5\%$ | $52.2\%$ |
> | Consistency | $86.4\%$ | $88.5\%$ | $88.7\%$ | $88.7\%$ |
>
> Q4: Regarding the implementation of AWR, it seems to be applied to each cluster individually.
>
> A4: Our method is scalable with respect to the number of clusters. Firstly, in our algorithm, we employ a soft clustering approach where each trajectory has a probability of belonging to each cluster. This partially mitigates the problem by allowing trajectories to contribute to multiple clusters. Secondly, policies of different styles share a common part of the network parameters, which helps in learning better stylized policies. Further details about the shared network structure can be found in the Appendix E.
>
> Q5: The paper omits a benchmark against "robust imitation of diverse behaviors".
>
> Q6: The paper could explore the potential of employing more advanced offline reinforcement learning algorithms in the second step of the methodology.
>
> A5,6: Thank you for bringing to our attention the benchmark "robust imitation of diverse behaviors" and the potential use of advanced offline RL algorithms in the second step of our methodology. We will definitely consider these suggestions for future improvements.

---

### Meta-Review · Area_Chair_kmrf · 2023-12-06

**Metareview:**

The paper proposes an offline RL method that extracts diverse policies from batched data. The reviewers praised the novelty of the solution, its potential for broad impact, and appreciated that the experiments were conducted on a real video game. There were concerns about comparisons against offline RL benchmarks, which the authors addressed through additional experiments. They also provided details of their experiments. Overall, this is a highly original paper, and the results were made more compelling following the rebuttals.

**Justification For Why Not Higher Score:**

While the idea is good, the initial version of the paper was rough around the edges. The results were improved following the rebuttals, but as I don't have access to the final version of the paper, I can't be sure it will be polished enough by the camera-ready deadline to warrant a spotlight.

**Justification For Why Not Lower Score:**

Unanimous accept.

---

### Decision · Program_Chairs · 2024-01-16

Accept (poster)